# Soil moisture: variable in space but redundant in time

Mirko Mälicke[1], Sibylle K. Hassler[1], Theresa Blume[2], Markus Weiler[3], and Erwin Zehe[1]

[1]Institute for Water and River Basin Management, Karlsruhe Institute of Technology (KIT), Germany
[2]GFZ German Research Centre for Geosciences, Section Hydrology, Germany
[3]Hydrology, Faculty of Environment and Natural Resources, University of Freiburg, Germany

**Correspondence:** Mirko Mälicke (mirko.maelicke@kit.edu)

**Abstract.**

Soil moisture at the catchment scale exhibits a huge spatial variability. This suggests that even a large amount of observation points would not be able to capture soil moisture variability.

We present a measure to capture the spatial dissimilarity and its change over time. Statistical dispersion among observation points is related to their distance to describe spatial patterns. We analyzed the temporal evolution and emergence of these patterns and use the Mean shift clustering algorithm to identify and analyze clusters. We found that soil moisture observations from the $19.4\,\mathrm{km}^2$ Colpach catchment in Luxembourg cluster in two fundamentally different states. On the one hand, we found rainfall-driven data clusters, usually characterized by strong relationships between dispersion and distance. Their spatial extent roughly matches the average hillslope length in the study area of about $500\,\mathrm{m}$. On the other hand, we found clusters covering the vegetation period. In drying and then dry soil conditions there is no particular spatial dependence in soil moisture patterns and the values are highly similar beyond hillslope scale.

By combining uncertainty propagation with information theory, we were able to calculate the information content of spatial similarity with respect to measurement uncertainty (when are patterns different outside of uncertainty margins?). We were able to prove that the spatial information contained in soil moisture observations is highly redundant (differences in spatial patterns over time are within the error margins). Thus, they can be compressed (all cluster members can be substituted by one representative member) to only a fragment of the original data volume without significant information loss.

Our most interesting finding is that even a few soil moisture time series bear a considerable amount of information about dynamic changes of soil moisture. We argue that distributed soil moisture sampling reflects an organized catchment state, where soil moisture variability is not random. Thus, only a small amount of observation points is necessary to capture soil moisture dynamics.

# 1 Introduction

Although soil water is by far the smallest fresh water stock on earth, it plays a key role in the functioning of terrestrial ecosystems. Soil moisture controls (preferential) infiltration and runoff generation and is a limiting factor for vegetation growth. Plant-available soil water affects the Bowen ratio i.e. the partitioning of net radiation energy in latent and sensible heat, and

last but not least it is an important control for soil respiration and related trace gas emissions. Technologies and experimental strategies to observe soil water dynamics across scales have been at the core of the hydrological research agenda for more than 20 years (Topp et al., 1982, 1984). Since these early studies published by Topp, spatially and temporally distributed Time Domain Reflectometry (TDR) and Frequency Domain Reflectometry (FDR) measurements have been widely used to characterize soil moisture dynamics at the transect (eg. Blume et al., 2009) , hillslope (eg. Starr et al., 2002; Brocca et al.,

2007) and catchment scale (eg. Western et al., 2004; Bronstert et al., 2012). A common conclusion for the catchment scale is that soil moisture exhibits pronounced spatial variability and distributed point sampling often don't yield representative data for the catchment (see eg. Zehe et al. (2010); Brocca et al. (2012) or numerous studies given in 2.2 of Vereecken et al., 2008).

Although large spatial variability seems to be a generic feature of soil moisture, there is also evidence that ranks of distributed soil moisture observations are largely stable in time as observed at the plot (Rolston et al., 1991; Zehe et al., 2010), hillslope

(Brocca et al., 2007; Blume et al., 2009; Brocca et al., 2009), and even catchment scale (Martínez-Fernández and Ceballos, 2003; Grayson et al., 1997). This rank stability, which is also often referred to as temporal stability (Vanderlinden et al., 2012), can i.e. be used to improve sensor networks (eg. Heathman et al., 2009) or select the most representative observation site in terms of soil moisture dynamics (eg. Teuling et al., 2006). In both cases rank stability assumes some kind of organization in the catchment, otherwise this representativity would not be observed.

Soil moisture dynamics have been subject to numerous review works (eg. Daly and Porporato, 2005; Vereecken et al., 2008). More specifically, the temporal stability of soil moisture was reviewed by Vanderlinden et al. (2012). The authors analyzed a large number of studies with respect to the controls on time stability of soil water content (TS SWC), but yet "the basic question about TS SWC and its controls remain unanswered. Moreover, the evidence found in literature with respect to TS SWC controls remains contradictory" (Vanderlinden et al., 2012, p.2 l.2ff). We want to contribute by proposing a method that

helps to understand how and when spatial soil moisture patterns are persistent.

Soil moisture responds to two main forcing regimes, namely rainfall driven wetting or radiation driven drying. The related controlling factors and processes differ strongly and operate at different spatial and temporal scales and the soil moisture pattern reflects thus the multitude of these influences (Bárdossy and Lehmann, 1998). Hence, we hypothesise that periods in which different controlling factors were dominant are reflected in fundamentally different soil moisture patterns. This can

manifest itself in changes in the spatial covariance structure (Lark, 2012; Schume et al., 2003), either in form of changing nugget to sill ratios (spatially explained variance) (Zehe et al., 2010) or state dependent variogram ranges (spatial extent of correlation) (Western et al., 2004). In a homogeneous, flat and non-vegetated landscape the soil moisture pattern shortly after a rainfall event would be the imprint of the precipitation pattern and provide predictive information about its spatial covariance. In contrast, in a heterogeneous landscape driven by spatially uniform block rain events, the spatial pattern of soil moisture

would be a largely stable imprint of different landscape properties controlling through-fall, infiltration as well as vertical and lateral soil water redistribution. Without further forcing, the spatial pattern will gradually dissipate due to soil water potential depletion and by lateral soil water flows. We therefore hypothesize that differences in soil moisture (across space) are higher shortly after a rainfall event and are dissipated afterwards.

Landscape heterogeneity is thus a perquisite for temporally persistent spatial patterns found in a set of soil moisture time series. While most catchments are strongly heterogeneous, it is striking how spatially organized they are (Dooge, 1986; Sivapalan, 2003; McDonnell et al., 2007; Zehe et al., 2014; Bras, 2015). Spatial organization manifests for instance through systematic and structured patterns of catchment properties, such as a catena. This might naturally lead to a systematic variability of those processes controlling wetting and drying of the soil. One approach to diagnose and model systematic variability is based on
the covariance between observations in relation to their separating distance (Burgess and Webster, 1980) and geo-statistical interpolation or simulations methods (Kitanidis and Vomvoris, 1983; Ly et al., 2011; Pool et al., 2015).

A spatial covariance function describes how linear statistical dependence of observations declines with increasing separating distance up to the distance of statistical independence. This is often expressed as experimental variogram. Geostatistics relies on several assumptions such as second order stationarity (see e.g. Lark (2012) or Burgess and Webster (1980)), which are
ultimately important for interpolation. Due to the above-mentioned dynamic nature of soil moisture observations, the most promising avenue for interpolation would be a spatio-temporal geostatistical modeling of our data (Ma, 2002; De Cesare et al., 2002; Ma, 2003; Snepvangers et al., 2003; Jost et al., 2005).

However, here we take a different avenue, as we do not intend to interpolate. One of our goals is to detect dynamic changes in the spatial soil moisture pattern. Following Sampson and Guttorp (1992) we relate the statistical dispersion of soil moisture
observations to their separating distance to characterize how their similarity and predictive information declines with this distance (see section 2). More specifically, we analyze temporal changes in the spatial dispersion of distributed soil moisture data and hypothesize that a grouping of the data is possible solely based on the changes in spatial dispersion. We want to find out whether typical patterns emerge in time, how those relate to the different forcing regimes and whether those patterns are recurrent in time. The latter is an indicator for predictability and (self) - organization in dynamic systems (Wendi and Marwan,
2018; Wendi et al., 2018).

Zehe et al. (2014) argued that spatial organization manifests through a similar hydrological functioning. This is in line with the idea of Wagener et al. (2007) on catchment classification, or the early idea of a geomorphological unit hydrograph (Rodríguez-Iturbe et al., 1979; Sivapalan et al., 2011; Patil and Stieglitz, 2012). Recently, Loritz et al. (2018) corroborated the idea of Zehe et al. (2014) and showed that hydrological similarity of discharge time series implies that they are redundant.
Redundancy in our context means that new observations (over time) do not add significant new information to the data set of spatial dispersion. Thus, they can be compressed without information loss (Weijs et al., 2013). This combination of compression rate and information loss is understood to be a measure of spatial organization in our work. More specifically, Loritz et al. (2018) showed that a set of 105 hillslope models yielded, despite their strong differences in topography, a strongly redundant runoff response. Using Shannon entropy (Shannon, 1948) Loritz et al. (2018) showed that the ensemble could be compressed

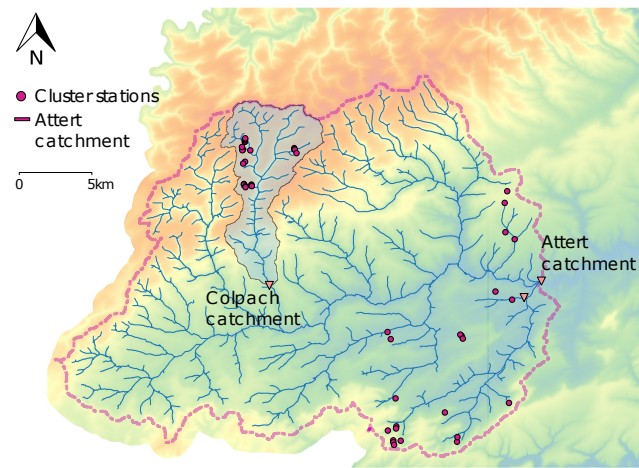

**Figure 1.** Attert experimental catchment in Luxembourg and Belgium. The purple dots show the sensor cluster stations installed during the CAOS project. Here we focus on those cluster stations within the Colpach catchment. Figure adapted after Loritz et al. (2017).

to a set of 6 to 8 typical hillslopes without performance loss. Here we adopt this idea and investigate the redundancy of patterns in spatially distributed soil moisture data along with their compressibility.

The core objective of this study is to provide evidence that distributed soil moisture time series provide, despite their strong spatial variability, representative information on soil moisture dynamics. More specifically, we test the following hypotheses:

5      – **H1:** Radiation-driven drying and rainfall-driven wetting leave different fingerprints in the soil moisture pattern.

     – **H2:** Both forcing regimes and their seasonal variability may be identified through temporal clustering of dispersion functions.

     – **H3:** Spatial dispersion is more pronounced during and shortly after rainfall driven wetting conditions.

     – **H4:** Soil moisture time series are redundant, which implies they are compressible without information loss. However,
10        the degree of compressibility is changing over time.

We test these hypotheses using a distributed soil moisture data set collected in the Colpach catchment in Luxembourg. In section 2 we give an overview of the study site and our method. The results section consists of three parts: spatial dispersion functions, temporal patterns in their emergence and some insights on generalization (or compressibility) of these functions, followed by a discussion and summary.

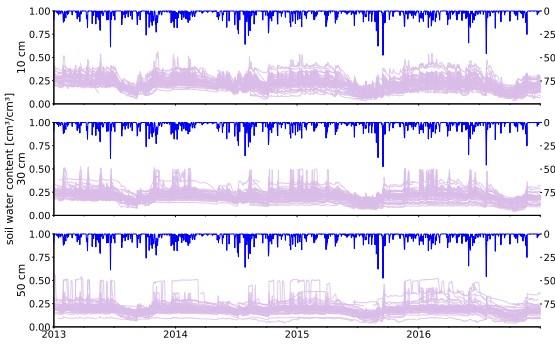

**Figure 2.** Soil moisture data overview. Soil moisture observations in 10 cm (top), 30 cm (middle) and 50 cm (bottom).

## 2   Methods

### 2.1   Study area and soil moisture data set

We base our analyses on the CAOS data set, which was collected in the Attert experimental watershed between 2012 and 2017 and is explained in Zehe et al. (2014). The Attert catchment is situated in western Luxembourg and Belgium (Figure 1). Mean monthly temperatures range from $18°$C in July to $0°$C in January. Mean annual precipitation is approximately 850 mm (Pfister et al., 2000). The catchment covers three geological formations, Devonian schists of the Ardennes massif in the northwest, a mixture of Triassic sandy marls in the center and a small area on Luxembourg Sandstone on the southern catchment border (Martinez-Carreras et al., 2012). The respective soils in the three areas are haplic Cambisols in the schist, different types of Stagnosols in the marls area and Arenosols in the sandstone (IUSS Working Group, 2006; Sprenger et al., 2016). The distinct differences in geology are also reflected in topography and land use. In the schist area, land use is mainly forest on steep slopes of the valleys, which intersect plateaus that are used for agriculture and pastures. The marls area has very gentle slopes and is mainly used for pastures and agriculture, while the sandstone area is forested on steep topography.

The experimental design is based on spatially distributed, clustered point measurements within replicated hillslopes. Typical hillslope lengths vary between 400 and 600 m showing maximum elevations of 50 to 100 m above stream level. For further details on the hillslopes we refer to Figure 6a in Loritz et al. (2017) and a detailed description in section 3.1.1 of the same publication. Sensor clusters were installed on hillslopes at the top, midslope and hill foot sector along the anticipated flow paths. Within each of those clusters, soil moisture was recorded in three profiles in 10, 30 and 50 cm depth using Decagon 5TE sensors. While the entire design was stratified to sample different geological settings (schist, marls, sandstone), different aspects and land use (deciduous forest and pasture), we focus here on those sensors installed in the Colpach catchment. In total we used 19 sensor cluster locations and thus 57 soil moisture profiles consisting of 171 time series.

Soil moisture in the 19.4 km$^2$ Colpach catchment exhibits high but temporally persistent spatial variability (Fig. 2). For each point in time a wide range of water content values can be observed across the catchment. The range of soil moisture observations is generally wider in winter than in summer. From visual inspection it seems that the heterogeneity in observations is not purely random but systematic as the measurements are rank stable over long periods. One has to note that the different cluster locations differ in is aspect, slope and landuse. From the data shown in figure 2, two sensors have been removed. Both measured in 50 cm and can be seen in the figure at the very bottom. Both recorded values close to or even below 0.1 cm$^3$cm$^{-3}$ for the whole period of four years. Additionally, the plateaus lasting for a couple of days at constant 0.5 cm$^3$cm$^{-3}$ in 50 cm and 30 cm were removed.

## 2.2 Dispersion of soil moisture observations as function of their distance

We focus on spatial patterns of soil moisture and how they change over time. For our analysis the data set was aggregated to mean daily soil moisture values $\theta$. Each time series is further aggregated using a moving window of one month as described by equation (1).

$$z_x(t) = \frac{\sum_t^{t+b} \theta_x}{b} \tag{1}$$

This is calculated for each observation location $x$ and time step $t = 1, 2, \ldots, (L-b)$, with a time series length of $L$ in days and a window size of $b = 30$ [1] .

To estimate the spatial dependence structure between observations, we relate their pairwise separation distance to a measure of pairwise similarity. Here, we further define the statistical spatial dispersion to be a measure of spatial similarity. We compare the empirical distribution of pairwise value differences at different distances. Statistically, a more dispersed empirical distribution is less well described by its mean value. Thus, observations taken at a specific distance are more similar in value, if they are less dispersed.

To estimate the dispersion, we use the Cressie-Hawkins estimator (Cressie and Hawkins, 1980). This estimator is more robust to extreme values and the contained power transformation handles skewed data better than estimators based on the arithmetic mean (Bárdossy and Kundzewicz, 1990; Cressie and Hawkins, 1980). The estimator is given by equation (2):

$$a_t(h) = \frac{1}{2} \left( \frac{1}{N(h)} \sum_{i,j} \sqrt{|z_t(x_i) - z_t(x_j)|} \right)^4 \left( 0.457 + \frac{0.494}{N(h)} + \frac{0.045}{N^2(h)} \right)^{-1} \tag{2}$$

for each moving window position $t$ with $z_t(x_{i,j})$ given by equation (1) for each pair of observation locations $x_i, x_j$. $h$ is the separating distance lag between these points pairs and $N(h)$ the number of points pairs formed at the given lag $h$. 10 classes

---

[1]We tested different window sizes, as we expect that different processes control the emergence of spatial dependence at different temporal scales. The chosen window size was most suitable to detect seasonal effects.

were formed with a maximum separation distance of 1200 m². The lag classes are not equidistant, but with a fixed $N(h)$ for all classes. This is further discussed in section 2.3.

## 2.3 Clustering of dispersion functions

We analyzed how and if meaningful spatial dispersion functions emerge and whether those converge into stable configurations. To tackle the hypotheses formulated in the introduction a clustering is applied to the dispersion functions derived for each window. The clustering algorithm should form groups of functions that are more similar to each other than to members of other clusters. The similarity between two dispersion functions is calculated by the Euclidean vector distance between the dispersion values forming the function. This distance is defined by equation (3):

$$d(\boldsymbol{u}, \boldsymbol{v}) = \sqrt{(\boldsymbol{u} - \boldsymbol{v})^2} \tag{3}$$

with $\boldsymbol{u}, \boldsymbol{v}$ being two dispersion function vectors. This is the Euclidean distance of two points in the (higher dimensional) value space of the dispersion function's distance lags. Two identical dispersion functions are represented by the same point in this value space and hence their distance is zero. Thus, distance lags are not equidistant, as this could lead to empty lag classes. Empty lag classes result in an undefined position in the value space, which has to be avoided. The clustering algorithm cannot use the number of clusters as a parameter, as this can hardly be determined a priori. One clustering algorithm meeting these requirements is the *Mean shift* algorithm (Fukunaga and Hostetler, 1975). The actual code implementation is taken from Pedregosa et al. (2011), which follows the Comaniciu and Meer (2002) variant of Mean shift. A detailed description of the Mean shift algorithm can be found in the appendix (see A).

## 2.4 Cluster compression based on the cluster centroids

The next step is to generate a representative dispersion function for each cluster. The straightforward representative function is the cluster centroid (the dispersion function closest to the point of highest cluster member density, see appendix A for a detailed explanation). All dispersion functions are calculated with the same parameters, including the maximum separating distance of 1200 m. At larger lags we found instances of declining dispersion values, because we then paired points located on different hillslopes, but otherwise in similar landscape units (i.e. same hillslope position or land use). To facilitate the comparison of the dispersion functions we decided to monotonize them. In geostatistics this is usually done through fitting of a theoretical variogram model to the experimental variogram, which assures monotony and positive definiteness. Here we do not force a specific shape by a fitting a model function. Instead we use the technique of monotonizing the cluster centroid as suggested by Hinterding (2003) using the PAVA-Algorithm (Barlow et al., 1972). The implementation is from Pedregosa et al. (2011). This way, the final compressed dispersion functions are monotonically increasing, while still reflecting the shape properties of the cluster members. If dispersion functions are monotonically increasing, they also provide information about the characteristic

---

[2] Observation point pairs further apart than 1200 m are most likely located on different hillslopes. These points might share similar soil, topographic and terrain aspect characteristics. Soil misture dynamics might thus be similar, although they are located at rather lage separating distances

length of the soil moisture pattern. Similar as for the semi-variogram in geostatistics this characteristic length corresponds the the lag distance where the dispersion function reaches its first local maximum.

We suggest that the number of clusters needed to represent all observed spatial dispersion functions over a calendar year can be used as a measure of spatial organization (fewer clusters needed means higher degree of organization, because dispersion functions are redundant in time). Additionally, it is insightful to judge the information loss that goes along with this compression, as a high compression with little information loss is understood as a manifestation of spatial and temporal organization of soil moisture dynamics.

In line with Loritz et al. (2018) we use the Shannon Entropy as measure for the compression without information loss. It requires treatment of the clusters as discrete probability density functions, which in turn implies a careful selection of an appropriate classification of the data. Motivated by Loritz et al. (2018), we use the uncertainty in the dispersion function as a minimal class size for this classification, as described in section 2.5.1.

## 2.5 Uncertainty propagation and compression quality

### 2.5.1 Uncertainty propagation

Soil moisture measurements have considerable measurement uncertainty of 1 - 3 $cm^3cm^{-3}$ as reported by manufacturers. For our uncertainty propagation we assume an absolute uncertainty/measurement error $\Delta\theta$ of 0.02 $cm^3cm^{-3}$.

Next we propagate these uncertainties into the dispersion functions and the distances among those. As we assume the measurement uncertainties to be statistically independent, we use the Gaussian uncertainty propagation to calculate error bands / margins . In a general form, for any function $f(z)$ and an absolute error $\Delta z$ the propagated error $\Delta f$ can be calculated. In our case $z$ is itself a function of $x$, the observation location, and the general form is given by equation (4).

$$\Delta f = \sqrt{\sum_{i=1}^{N} \left( \frac{\partial f}{\partial z(x_i)} \Delta z(x_i) \right)^2} \tag{4}$$

To apply equation (4) for our method, the measurement uncertainty $\Delta\theta$ is propagated into the dispersion estimator given by equation (2). The dispersion estimator is derived with respect to $z(x)$ and following equation (1), the uncertainty in $z(x)$, $\Delta z$, is denoted as $\Delta z = \Delta\theta = 0.02cm^3cm^-3$. Then, with given $\Delta z$, we can propagate the uncertainty into the dispersion function. As the dispersion function is a function of the spatial lag $h$, we need to propagate the uncertainty $\Delta a$ (uncertainty of dispersion estimator) for each value of $h$. At the same time, following equation (2), for each $h$, $z(x_i)-z(x_i+h)$ is a fixed set of point pairs. Instead of propagating uncertainty through equation (2), we can substitute $z(x_i) - z(x_i + h)$ by $\delta$, the pairwise differences, for each value of $h$. The uncertainty $\Delta\delta$ is given by equation (5)

$$\Delta\delta = \sqrt{\Delta z_i^2 + \Delta z_{i+h}^2} = \sqrt{2}\Delta z \tag{5}$$

The uncertainty of dispersion $\Delta a$ is then defined by equation (6):

$$\delta a = \frac{\partial a}{\partial \delta} \Delta \delta \tag{6}$$

$$= 2c \left( \frac{1}{N} \sum_{i=1}^{N} (|\delta_i|)^{\frac{1}{2}} \right)^3 \cdot \frac{1}{N} \left( \sum_{i=1}^{N} |\delta_i|^{-}1 \right)^{\frac{1}{2}} \cdot \Delta \delta \tag{7}$$

where the factors from equation (2) that stay constant in the derivative are denoted as $c$ and defined in equation (8). In line with equation (2) $N$ is the number of observation pairs available for a given lag class $h$ and therefore constant for a single calculation. $\Delta \delta$ and $\delta$ are the substitutes for $z$, as described above (see equation (5)).

$$c = \frac{1}{2} * (0.457 + \frac{1}{N} + \frac{0.045}{N^2})^{-1} \tag{8}$$

The last step is to propagate the uncertainty into the distance function as defined in equation (3). The Euclidean distance is used as a measure for proximity by Mean shift, as it groups dispersion functions at short distances together (for more details see appendix section A). At the same time, we use the uncertainty propagated into the Euclidean distance between two dispersion functions to assess compression quality (as further described in section 2.5.2). Following equation (4) the propagated uncertainty $\Delta D$ can be calculated by the derivative of equation 3 with respect to each of the vectors multiplied by the corresponding value of $\Delta a$, which results in equation (9):

$$\Delta d_{\boldsymbol{u},\boldsymbol{v}} = \sqrt{\left( \frac{\delta d}{\delta \boldsymbol{u}} \boldsymbol{\Delta u} \right)^2 + \left( \frac{\delta d}{\delta \boldsymbol{u}} \boldsymbol{\Delta u} \right)^2} \tag{9}$$

$$= \sqrt{\frac{1}{2} \sum_{i=1}^{n} (|\boldsymbol{u} - \boldsymbol{v}|^{\frac{1}{2}}) \sum_{i=1}^{n} ((2(|\boldsymbol{u}_i - \boldsymbol{v}_i|)\Delta \boldsymbol{u}_i)^2 + (2(|\boldsymbol{u}_i - \boldsymbol{v}_i|)\Delta \boldsymbol{v}_i)^2)} \tag{10}$$

Where $\boldsymbol{u}, \boldsymbol{v}$ are two spatial dispersion function vectors as defined and used in equation (3). $\boldsymbol{\Delta u}, \boldsymbol{\Delta v}$ are the vectors of uncertainties for $\boldsymbol{u}, \boldsymbol{v}$, where $\Delta v_i$ is the uncertainty propagated into the $i^{th}$ lag class as shown in equation (6). $n$ is the number of lag classes and thus the length of each vector $\boldsymbol{u}, \boldsymbol{v}, \boldsymbol{\Delta u}, \boldsymbol{\Delta v}$.

Equation (9) is applied to all possible combinations of dispersion functions $\boldsymbol{u}, \boldsymbol{v}$ to get all possible uncertainties in dispersion function distances.

### 2.5.2 Compression quality

The Shannon entropy (Shannon, 1948) of all pairwise dispersion function distances is used as measure for information content. The Shannon entropy of a discrete probability density function of states (patterns in this case) is maximized for the uniform distribution. It corresponds to the number of yes/no questions one has to ask to determine the state of a system. The minimum entropy is zero, which corresponds to the deterministic case that the system state is always known. A common way to define spatial organisation of a physical system is through its distance from the maximum entropy state (Kondepudi and Prigogine,

1998; Kleidon, 2012). The deviation of the entropy of the dispersion functions in a cluster from its maximum value is thus a measure for their redundancy and thus similarity.

For a discrete frequency distribution of $n$ bins, the information entropy $H$ is defined as:

$$H = -\sum_n p_n log_2(p_n) \tag{11}$$

Where $p_n$ is the relative probability of the $n$th bin. $H$ is calculated for each depth in each year individually to compare the information content across years and depths. Note that the term *bin* is also used in literature to refer to the *binning* of pairwise data, e.g. in geo-statistics. For this kind of binning, although technically the same thing, we used the term *lag classes* here, to distinguish from the binning as shown in equation (11). Thus, when we write *bin* or *binning* we refer to the classification of distances between dispersion functions, not observation points.

To assure comparability we use one binning for all calculations of $H$ (across years and depths). To achieve this, all pairwise distances between all spatial dispersion functions of all four years in all three depths are calculated. The discrete frequency distribution is formed from 0 up to the global maximum distance (between two dispersion functions) calculated using equation (3). The bins are formed equidistant using a width of the maximum function distance that still lies within the error margins calculated using equation (9). Thus, the information content of the spatial heterogeneity is calculated with respect to the

expected uncertainties. This way we can be sure to distinguish exclusively those spatial dispersion functions that lie outside of the error margins.

The Kullback-Leibler divergence (Kullback and Leibler, 1951) is a measure for the difference between two empirical, discrete probability distributions. Usually, one distribution is considered to be the population and the other one a sample from it. The Kullback-Leibler divergence $D_{KL}$ then quantifies the uncertainty introduced (e.g. in an statistical model) using a sample

as a substitute for the population.

We use the Kullback-Leibler divergence to measure and quantify the information loss due to compression. To compress the series of dispersion functions, each cluster member is expressed by its centroid function. Now, we need to calculate the amount of information lost in this process. To calculate the mean information content of the compressed series each cluster member is substituted by the respective cluster centroid. This substitution is obviously not a compression in a technical sense,

but necessary to calculate the Kullback-Leibler divergence. Then a frequency distribution for compressed series $X$ and the uncompressed series $Y$ can be calculated. The Kullback-Leibler divergence $D_{KL}$ of $X, Y$ is given in equation (12):

$$D_{KL}(X,Y) = H(X||Y) - H(Y) \tag{12}$$

Where $H(X||Y)$ is the cross entropy of $X$ and $Y$ and defined by equation (13):

$$H(X||Y) = \sum_{x \in X} p(x) * log_2 p(y) \tag{13}$$

Where $p(x)$ is the empirical non-exceedance probability of the frequency distribution $X$ and $p(y)$ of $Y$, respectively.

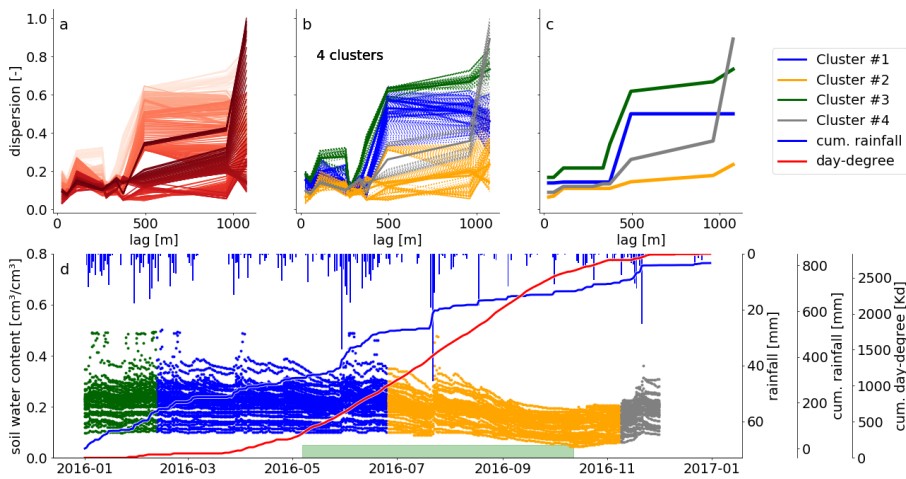

**Figure 3.** Spatial dispersion functions in 30 cm for 2016 based on on a window size of 30 days.

**a):** Spatial dispersion function for each position of the moving window. The red color saturation is indicating the window position. The darker the red the later in the year.

**b):** The same dispersion functions as presented in a). Here the color indicates cluster membership as identified by the Mean shift algorithm.

**c):** Compressed spatial dispersion information represented by corrected cluster centroids. The colors match the clusters as presented in b).

**d):** Soil moisture time series of 2016 in 30 cm depth. The colors identify the cluster membership of the spatial dispersion function of the current window location and is matching the colors in b) and c). The bars on the top show the daily precipitation sums. The solid blue line is the cumulative daily precipitation sum and the red line the cumulative sum of all mean daily temperatures $> 5°C$. The green bar marks the assumed vegetation period. It covers the dates where the cumulative day-degree sum is $> 15\% \ \& < 90\%$ of the maximum.

## 3  Results

### 3.1  Dispersion functions over time

Figure 3 a) shows the spatial dispersion functions for all moving window positions in 2016 for the 30 cm sensors. The position of the moving window in time can be retraced by the line color, darker red means a later Julian day. Each of the spatial
5  dispersion functions relates the dispersion for all pairwise observations to their separating distance in the corresponding lag class. Dispersion increases with separating distance, as small values correspond to observations which have similar values while large values suggest the opposite. As expected, the dispersion is a suitable metric for similarity/dependency of observations.

The spatial dispersion functions take several distinct shapes with each of these shapes occurring during a certain period in time. More specifically from Figure 3 a) one can identify groups of functions of similar reds plotting close to each other.
10  Dispersion functions of similar red saturation, which reflects proximity in time, are also similar in shape, and this in turn reflects similar spatial patterns. Similar dispersion functions were grouped using the Mean shift clustering algorithm (Fig. 3 b); here, the color indicates the cluster membership.

To provide further insight on the temporal occurrence of cluster members, we colored the soil moisture time series according to the color codes of the identified clusters (Fig. 3 d). The blue parts of the soil moisture time series were classified into Cluster #1, while the the orange was classified into Cluster #2. Note that cluster memberships are constant for long periods of time, which means that also the soil moisture patterns are persistent over these periods. Exact cluster lifespans can be found in table B1. We could identify four clusters in 30 cm, with the orange cluster roughly occurring during the vegetation period and the other three the remaining time of the year. As new observations did not change the patterns during these periods, they were redundant in time.

As the spatial dispersion functions in the presented example are redundant in time, we compressed the information by replacing the dispersion function within one cluster by the cluster centroid. All four representative functions shown in Figure 3 c) exhibit increasing dispersion with separating distance. For the blue and green cluster this happens step-wise at a characteristic distance of 500 m. That reminds us of a Gaussian variogram, which can also show a step-wise characteristic. The small grey cluster shows an increase at 500 and another one at 1000 m separating distance. In contrast the orange cluster, however, shows a only a gentle increase with distance.

In the vegetation period observations are similar even at large separating distances. Interestingly, dispersion functions in the orange cluster start with small values that only gently increase with separating distance. That means soil moisture becomes more homogeneous. Outside of the vegetation period, different spatial patterns can be observed, with increasing dissimilarity with separating distances. The part of the blue cluster overlapping with vegetation period shows still higher soil moisture values. The transition to the orange cluster sets in as the soil moisture drops (Fig. 3 d). This suggests that vegetation influences, such as root water uptake, smooth out variability in soil water content, leading to a more homogeneous pattern in space, as further discussed in section 4.3.

## 3.2 Dispersion time series as a function of depth

Figure 4 shows the time series of the dispersion functions for all depths. Note that the coloring between the sub-figure is arbitrary, due to Mean shift, that means there is no connection between the orange cluster between the three figures.

In comparison to the dispersion functions in 30 cm (Fig. 4 b) the soil moisture signal in 10 cm (Fig. 4 a) is more variable in time. A look at the centroid of the orange cluster (Fig. 4 d) reveals a higher spatial heterogeneity in winter and spring at large separating distances. At the same time the observations get spatially more homogeneous in summer, particularly when the blue cluster emerges, i.e. the dispersion at large lags decreases significantly. We can still find a summer-recession cluster in 10 cm, but compared to the depth of 30 cm we also find this spatial footprint of continuous drying earlier in the year around May. This is likely due to a higher sensitivity to rising temperatures. Note that during May there was only little rainfall and the soil moisture is already declining. This blue cluster shows very small dispersion values for all separating distance classes (Fig. 4 d), just as the orange cluster in 30 cm depth.

The green clusters emerge with strong rainfall events after longer previous dry spells (Fig. 4, a,d). We would have expected a third occurrence at the beginning of August, but the soil may already be too dry to bear a detectable dependency on separating distance (Remember that the blue cluster does not show increasing dispersion with distance).

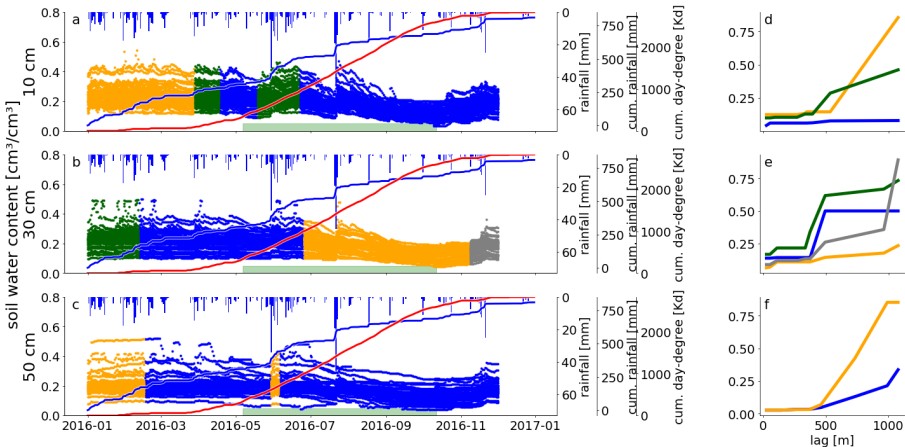

**Figure 4.** Soil moisture time-series of 2016 in all three depths with respective cluster centroids. The three rows show the data from 10 cm (a), 30 cm (b) and 50 cm (c). The colors indicate the cluster membership of the corresponding dispersion function of the respective window position. The green bar marks the assumed vegetation period. It covers the dates where the cumulative day-degree sum is > 15% & < 90% of the maximum. The cluster centroids for each depth is shown in d, e and f.

Observations at 50 cm depth show a clear spatial dependency throughout the whole year. We cannot identify a summer cluster, Mean shift yielded two clusters and rainfall forcing does not have a clear influence on their occurrence or transition. The two 50 cm dispersion functions (Fig. 4 f) show a clear dependence on distance, but they differ in their dispersion value at large lags. In 10 cm and 30 cm we found dispersion functions of fundamentally different shape, like the flat, blue function
5    (Fig. 4, d) or the step-wise blue and green functions (Fig. 4 e). In 50 cm depth the characteristic length is 500 m and the blue cluster persists throughout most of the year (282 days, see table B1). The orange cluster occurs during the cool and wet start of the year, showing a larger dispersion and thus stronger dissimilarity at larger lags (Fig. 4 f). Interestingly this cluster occurs again in early June after an intense rainfall period. However, a similar rainfall period in August does not trigger the emergence of this orange cluster as the top soil above 50 cm is so dry, that even this strong wetting signal does not reach the depth of 50
10    cm (Fig. 4 c). This behavior reveals the low pass behavior of the top soil, which causes a strong decoupling of the soil moisture pattern in 50 cm depth from event scale changes.

### 3.3    Recurring spatial dispersion over the years

Table 1 summarizes the most important features of the clustering for all observation depths. Soil moisture patterns and their clustering appear generally to be clearer for 2015 and 2016. The vegetation period is more often characterized by a typical
15    cluster and dispersion functions more often reveal a clear spatial dependency. In some cases (10 cm, 2013 and 2014) no spatial dependency of dispersion functions could be observed throughout the whole year. Less clusters were formed in 2015 and 2016. Note that annual rainfall sums were higher in 2013 and 2014, while 2015 and 2016 had significantly more precipitation in the first half of the year followed by a dry summer (compared to 2013 and 2014).

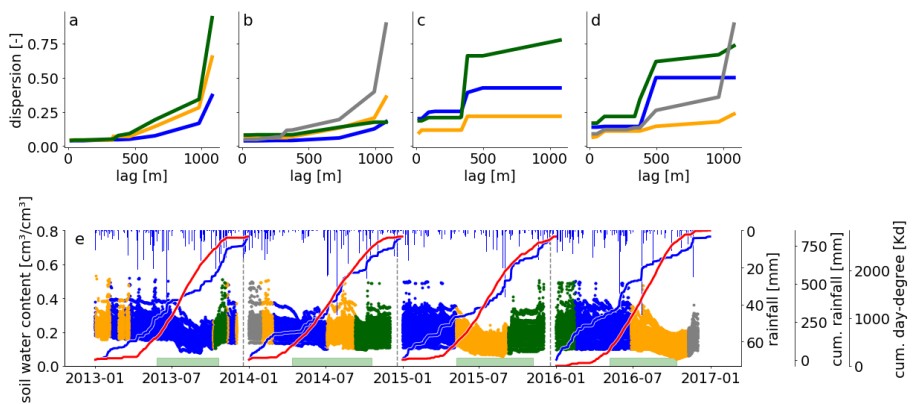

**Figure 5.** Soil moisture timeseries of all years in 30 cm depth (e) and the respective cluster centroids (a-d). The colors of the soil moisture data indicate the cluster membership of the corresponding dispersion function of the respective window position and correspond with the color of the cluster centroid (in a-d). The cumulative rainfalls (blue) and cumulative temperature sums (red) are shown for each year individually. The green bar marks the assumed vegetation period. It covers the dates where the cumulative day-degree sum is > 15% & < 90% of the maximum.

**Table 1.** Qualitative description of method success in all years and depths. The results from years other than 2016 and all depths were inspected visually and are summarized here for sake of completeness. The first three columns identify the year, sensor depth and the number of clusters found by Mean shift. The remaining three columns state if specific features existed in the given result. *Vegetation period* marks whether or not the vegetation period was characterized by a single, or two, clusters. *Spatial structure*: Does a dependency of dispersion on distance exist **outside** the vegetation period? *rainfall transition*: Were cluster transitions accompanied by a rainfall event in close (temporal) proximity? This feature is marked 'yes' if it was more often the case than it was not.

| Year | depth | # of clusters | vegetation period | spatial structure | rainfall transition |
|------|-------|---------------|-------------------|-------------------|---------------------|
| 2013 | 10 | 4 | yes | no | no |
| 2013 | 30 | 3 | no | yes | yes |
| 2013 | 50 | 6 | no | yes | yes |
| 2014 | 10 | 3 | yes | no | yes |
| 2014 | 30 | 4 | no | yes | no |
| 2014 | 50 | 5 | no | yes | yes |
| 2015 | 10 | 3 | yes | yes | no |
| 2015 | 30 | 3 | yes | yes | yes |
| 2015 | 50 | 3 | no | yes | no |
| 2016 | 10 | 3 | no | yes | yes |
| 2016 | 30 | 4 | yes | yes | yes |
| 2016 | 50 | 2 | no | yes | no |

To further illuminate inter annual changes in soil moisture patterns we present the time series of cluster memberships for the sensors in 30 cm for the entire monitoring period in Figure 5. From this example it becomes obvious that patterns are recurring. 2013 and 2014 cluster centroids look different from the following two years. Dispersion values increase with distance in all centroids in 2013 and 2014, while 2015 and 2016 show a sudden increase at 400 - 500 m (Fig. 5 a-d). 2015 and 2016 are

segmented by Mean shift in a similar way, and cluster centroids reveal that the green clusters in both years are actually the same. This green cluster emerges with the occurrence of the largest rainfall event in the observation period and lasts for around 5 months. All dispersion functions within this cluster look nearly identical (see Appendix, Fig. C2, b). Similar observations can be made between 2014 and 2015. Here, the green and blue cluster seem to be an inter-annual cluster. However, in contrast to to 2015/2016 the dispersion functions here are of different shape (see Appendix, Fig. C1, b). Hence, the cluster transition

indicated between 2014 and 2015 is indeed a real transition. When looking at cluster memberships throughout the whole period, the division into calendar years is rather meaningless, while the division in hydrological years is much more appropriate, as it is reflected by the cluster membership and its changes.

Distinct summer recessions in soil moisture are only identified in 2015 and 2016. Evapotranspiration (indicated by the cumulative temperature curves in Fig. 5 e) is dominating over rainfall input (blue sum curve) in the soil moisture signal. Mean

shift could identify a significantly distinct spatial dependency in dispersion, as shown by the two orange centroids in figure 5 c and d). They are both distinct from the other centroids in the same period by showing only a gentle increase in dispersion. A likely reason for the absence of a distinct summer recession in 2014 is the rather wet and cold spring and summer, as can be seen from the steep cumulative rainfall curve during that period (Fig. 5 e). In 2013 this identification did not work. Possible reasons are provided in the discussion section 4.5.

## 3.4   Redundant spatial dispersion functions

We calculated the Shannon entropy for all soil moisture time series for all years and depths (Table 2). As explained in section 2.5.2 this reflects the intrinsic uncertainty of the clusters. Most entropy values are within a range of $1 < H < 2.5$. The maximum possible entropy for a uniform distribution of the used binning is $3.55$. The Kullback-Leibler divergence $D_{KL}$ is a measure for the information loss due to the compression of the cluster onto the centroid dispersion function. In the overwhelming majority of

the cases, the information loss is one magnitude smaller than the intrinsic uncertainty and ranges between $0.01 < D_{KL} < 0.4$. Hence, the information loss due to compression is negligible. There is one exception in 2016 (50 cm).

The clusters obtained in 30 cm for the year 2016 (compare 3.1) showed an entropy of 1.44. Compared to this value, the Kullback-Leibler divergence caused by compression of only 0.02 is small, if not negligible. The last column of table 2 relates $D_{KL}$ to the overall uncertainty. It contributes less than one third in almost all cases (2016, 50 cm is the only exception). In the

majority of the cases it does not contribute more than 20%.

According to equation (11) the Shannon entropy is derived from an discrete, empirical probability distribution. As it is calculated using the binary logarithm, $2^H$ gives the amount of discriminable states in this discrete distribution. This number of states is deemed to be a reasonable upper limit for the number of clusters for Mean shift. A higher number of clusters than $2^H$

**Table 2.** Information content and information loss due to compression. The information content is given as Shannon entropy $H$, which is the expectation value of information in information theory. $2^H$ gives the number of distinct states the underlying distribution can resolve. The information loss after compression is given by the Kullback-Leibler divergence $D_{KL}$ between the compressed and uncompressed series of dispersion functions. The last column relates $D_{KL}$ to $H$

| Year | Depth | # of clusters | $H$ [bit] | $2^H$ | $D_{KL}$ [bit] | $D_{KL} * (H + D_{KL})^{-1}$ |
|------|-------|---------------|-----------|-------|----------------|------------------------------|
| 2013 | 10 cm | 4 | 0.97 | 1.95 | 0.44 | 0.31 |
| 2013 | 30 cm | 3 | 1.49 | 2.81 | 0.06 | 0.04 |
| 2013 | 50 cm | 6 | 2.0 | 3.99 | 0.13 | 0.06 |
| 2014 | 10 cm | 3 | 1.35 | 2.55 | 0.22 | 0.14 |
| 2014 | 30 cm | 4 | 1.57 | 2.97 | 0.3 | 0.16 |
| 2014 | 50 cm | 5 | 2.44 | 5.42 | 0.28 | 0.1 |
| 2015 | 10 cm | 3 | 1.87 | 3.67 | 0.18 | 0.09 |
| 2015 | 30 cm | 3 | 1.18 | 2.26 | 0.09 | 0.07 |
| 2015 | 50 cm | 3 | 2.39 | 5.24 | 0.9 | 0.27 |
| 2016 | 10 cm | 3 | 2.49 | 5.62 | 0.76 | 0.23 |
| 2016 | 30 cm | 4 | 1.44 | 2.71 | 0.02 | 0.02 |
| 2016 | 50 cm | 2 | 3.21 | 9.27 | 2.5 | 0.44 |

appears meaningless, and this assures that only those clusters are separated, which are separated by a distance large than the margin of uncertainty.

## 4   Discussion

In line with our central hypothesis **H1** - that radiation-driven drying and rainfall-driven wetting leave different fingerprints
5  in the soil moisture pattern which manifests in temporal changes in the dispersion functions - we found strong evidence that soil water dynamics is organized in space and time. Our findings reveal that this organization is not static but exhibits dynamic changes which are closely related to seasonal changes in forcing regimes. A direct consequence is that soil moisture observations are quite predictable in time despite their strong spatial heterogeneity. This is in line with conclusions of e.g. Mittelbach and Seneviratne (2012) or Teuling et al. (2006), who also found characteristic spatial patterns to persist in time.
10  We used the statistical dispersion of soil moisture observations in dependence of their separating distance to describe spatial patterns. The vector distance of these dispersion functions was used to cluster them. As measure for the degree of organization we used the information loss that goes along with the compression of the entire cluster, ie the replacement of the cluster by the most representative cluster member. Here we found that this compression adds negligible uncertainty compared to the intrinsic uncertainty, caused by propagation of measurements uncertainties. We thus conclude that soil moisture is heterogeneous, but
15  temporally persistent over several months.

In the following we will discuss our main findings that similarity in space leads to dynamic similarity in time, the way we utilized the measurement uncertainty to determine the information content and how two different processes forcing soil moisture dynamics induce two fundamentally different spatial pattern.

## 4.1 Spatial similarity persist in time

We related the dispersion of pairwise point observations to their separating distance. For brevity and due to their shape we called these relationships *dispersion functions*. We emphasize that this term is not meant in a strict sense and no mathematical functional relationship, analogous to a theoretical model, has been fitted to the experimental dispersion functions. Despite the fact that the presented functions are empirical, they show clear, recurrent shapes on many occasions.

We found spatial similarity to persist in time. This is reflected in the temporal stability in cluster membership. In line
with **H2** - that both forcing regimes and their seasonal variability may be identified through temporal clustering of dispersion functions - the results (Fig. 3, 5, 4) provided evidence that similar dispersion functions emerge in fact very closely in time. Generally they appeared in continuous periods or blocks in time and their changes coincided with changes or a switch in the forcing regimes. In case we can relate the emergence of such a cluster more quantitatively to the nature and strength of a specific forcing event/process, we can analyze for how long this event/process imprints the spatial pattern of soil moisture
observations. Or in other words: we can analyze how long a catchment state *remembers* a disturbance. However, an attempt to relate cluster transitions to rainfall sums and frequencies within the respective moving windows (see Fig. B1) did not yield clear dependencies.

Although cluster memberships occur in temporally continuous blocks in all depths throughout all years, for a few cases we could not relate their emergence to distinct changes in forcing. This implies that **H2** needs to be partly revised.
Dispersion functions in 50 cm show a clear spatial dependency throughout the year, with distinct differences within and outside the vegetation period. In 50 cm of 2016 this is different. We find essentially two clusters that do not separate the data series by vegetation period. The shape of the two centroids (Fig. 4, f) is similar, only at large distances they differ in value. That means, from orange to blue cluster observations became more similar at large separating distances. Heavy rainfall disturbs this pattern leading to stronger dissimilarity at larger distances and that pattern lasted for a couple of weeks. Then,
evapotranspiration driven drying smooths out soil moisture variability and during a similarly strong rainfall event in summer, the cluster can not emerge again as the soil is already too dry. The soil acts as a low pass filter here, which filters out any change in state above a specific frequency. This happens mainly due to dispersion of the infiltrating and percolating water through the soil, or due to storage in the soil matrix. By the time it reaches the deep layers, spatial differences are eliminated. This kind of behaviour is well known and was already reported in the early 90s (Entekhabi et al., 1992; Wu et al., 2002).
More recently (Rosenbaum et al., 2012) "found large variations in spatial soil moisture patterns in the topsoil, mostly related to meteorological forcing. In the subsoil, temporal dynamics were diminished due to soil water redistribution processes and root water uptake". In the same year, Takagi and Lin (2012) analyzed a dataset of 106 locations in a forested catchment in the US for spatial organization in soil moisture patterns. They found a seasonal change in more shallow depths (30 cm), controlled

by rainfall and evapotranspiration. In deeper depths patterns became more temporally persistent. All these findings are in line with our results and conclusions.

Mittelbach and Seneviratne (2012) decomposed a long term (15 months) soil moisture time series into time-invariant and dynamic contributions to the spatial variance. Their dataset spanned 14 sites from Switzerland at a clearly different scale (150 x 210 km). The study quantified the time-invariant contribution on average to 94%, which leads to "a smaller spatial variability of the temporal dynamics than possibly inferred from the spatial variability of the mean soil moisture" (Mittelbach and Seneviratne, 2012, p.2177 L.14ff). This is comparable to the instances, where we find long lasting clusters while the absolute soil moisture changes considerably (e.g. Fig 3 d), early April or mid of July).

## 4.2  Uncertainty analysis

We related the evaluation of compression quality directly to the measurement uncertainty. This was achieved by Gaussian error propagation of measurement uncertainty into the dispersion functions and their distances. The latter allowed definition of a minimum separable vector distance between two dispersion functions that are different with respect to the error margin. We based the bin width for calculating the Shannon entropy on this minimum distance, because this assured that the Shannon entropy gives the information content of each cluster *with respect to the uncertainty*. On this basis it was possible to assess compression quality not only by the number of meaningful clusters found, but also based on the information lost due to compression with respect to uncertainty.

In line with **H4** spatial patterns of soil moisture were found to be persistent over weeks, if not months. In many instances we found only two to four clusters within one year and compression was possible with small if not negligible information loss. That means, during one cluster period an entire set of dispersion functions does not contain substantially more information than the centroid function. Hence, the whole cluster can be represented by only the centroid function. We conclude that this is a manifestation of a strongly organized state which persist for a considerable time, as most observations were redundant during these periods.

Teuling et al. (2006) concluded that picking a random soil moisture observation location and deriving the temporal dynamics from this single sensor is more accurate than using the spatial mean of many soil moisture time series. This conclusion was true for all three datasets they tested (Teuling et al., 2006). This representativity of a single sensor to our understanding a manifestation of a persistent spatial pattern in soil moisture dynamics, which also enables us to compress clusters without information loss.

From equation (11) it can be seen, that the Shannon entropy changes substantially with the binning. Therefore, it is of crucial importance to define a meaningful binning based on objective criteria. We suggest that only a discrimination into bins larger than the error margins makes sense, because smaller differences cannot be resolved based on the precision of the sensors. For the application presented in this work, this is important because otherwise one could not compare the compression quality between depths or years, as different binnings lead to different Shannon entropy values, even for the same data. Hence, it would be difficult to analyze effects or differences of spatial dispersion in depth or over the years. We thus conclude that the Shannon entropy should only be used if the measurement uncertainties of the data are properly propagated.

We provided an example of how the quality of a compression can be assessed. Instead of considering the number of clusters (compression rate) only, we linked the compression rate to the resulting information loss. We could show that in the majority of the cases substantial compression rates could be achieved, which are accompanied by negligible information losses. We thus suggest that the trade off between compression rate and information loss should be used as compression quality measure.

## 4.3   Different dominant processes lead to different patterns

Outside of the vegetation period, we found a recurring picture of spatial dispersion functions with characteristic lengths clearly smaller than the typical extent of hillslopes. Dispersion functions were calculated in three depths for every day throughout four years. In most cases there is an characteristic length at which the dispersion function shows a sudden rise in dispersion. For spatial lags smaller than this distance the dispersion is usually very small. Higher lags show much higher and more variable dispersion values. This characteristic length is approx. 500 m. This corresponds to a common hillslope length for the Colpach catchment. During the vegetation period variability at large separating distance was smoothed out. Dispersion was low also at large distances suggesting similarity even at distances larger than the typical slope length. We thus conclude that there is dependence of the dispersion on the rainfall pattern, which is reflected in the dispersion function's shape and characteristic length. This confirms **H2** and suggest that vegetation is a possible dominant factor in smoothing out soil moisture variability. A similar conclusion is drawn by Meyles et al. (2003), who identified 'preferred states in soil moisture' (Grayson et al., 1997; Western and Grayson, 1998; Western et al., 1999) and could relate the state transition to a significant change in the characteristic length of their geostatistical analysis. We generally found more than two clusters, but we still consider these results to be comparable. Most of the clusters identified during vegetation period are more similar to each other than to the clusters outside of the vegetation period (and vice versa). This can be related to the 'wet' and 'dry' state in Meyles et al. (2003). Although conducted in a very different climate McNamara et al. (2005) also widened the separation of two preferred states into five which they found to be explanatory for runoff generation. Interestingly they found the seasonal interplay of precipitation and evapotranspiration responsible for transitions between states. Vanderlinden et al. (2012) further references Gómez-Plaza et al. (2001) as an example study, which identified vegetation as the dominant factor. Plant root activity is changing the temporal stability of soil moisture in the upper 20 cm of the soil considerably.

Outside the vegetation period we observed multiple cluster transitions. Although more than one cluster was identified, the clusters were more similar in shape to each other, than to the clusters in the 'dry' summer period. In many cases these cluster transitions coincided with a shift in rainfall regimes. Either the first stronger rainfall event after a longer period without rainfall sets in, or one of the heaviest rainfall events of that year occurs. There are also instances with recurring clusters that develop more than once (eg. Fig. 3, 4 a, 4 c, 5 e). As these periods are controlled by rainfall either different rainfall patterns or different hydrological processes are dominating. Depending on antecedent wetness, rainfall amounts and rainfall intensity, infiltration and subsurface flow processes can change and thus also alter the soil moisture pattern. Although this may only be a coincidence, we found the green cluster in 2016 (Fig. 3) to form with strong rainfall input setting in after a period of little rainfall. Similar observations can be made for other years, unfortunately not in all cases. Consequently, we can neither confirm nor reject **H3** - that spatial dispersion is more pronounced during and shortly after rainfall driven wetting conditions.

Many other works also tried to link soil moisture pattern to forcing. Teuling and Troch (2005) report for soil moisture measurements taken on an agricultural field in Belgium, that the first rainfall events in the late growing season even out the variability, which arose due to heterogeneous transpiration. Although soil moisture pattern became more homogeneous in summer in our case, we also suspect rainfall events after the vegetation period to be responsible for cluster transitions. Similarly,

Albertson and Montaldo (2003) present a set of examples of modelled experiments, in which precipitation is consistently 'producing' variability in soil moisture dynamics and transpiration is reducing variability. The question of how spatial patterns or their variability change is also contradictory in the literature. Vanderlinden et al. (2012) present two studies in their review. Both investigated the variability of time persistent soil moisture patterns over depth. While Pachepsky et al. (2005) found no difference in depth, Choi and Jacobs (2011) reported a decrease of variability with depth. During the vegetation period no spatial

dependence is detectable. For the vegetation period, we found usually only one or at maximum two clusters (Table 2). These clusters are characterized by showing no dependence of dispersion on separating distance. That means, evapotranspiration forcing the system to drier states is doing this in a (spatially) homogeneous manner. Dispersion is not only low when the catchment is dry, it is also low while the system is drying. Similar observations have been reported for the Tarrawarra catchment in Australia (Grayson et al., 1997; Western and Grayson, 1998; Western et al., 1999). Although these works focused on the

relation of spatial organization to topographic indices, no spatial correlation of soil moisture observations could be found for the dry period. This is comparable to our to our findings about dispersion functions during the vegetation period. It has to be noted that the lowest soil moisture values, i.e. residual moisture, are only observed for very short periods in time. At residual soil moisture all sensors show essentially the same absolute value (which leads to small dispersion as well).

We conclude that cluster transitions were often triggered by rainfall events. Not each of the strongest rainfall events caused

a cluster transition and not each cluster transition could be related to a rainfall sums or frequency within the window of the transition. The characteristics provided in appendix B provide a good starting point, but further investigations on the rainfall events, their spatial characteristics and relation to the moisture state are needed.

## 4.4   Mean shift as a diagnostic tool

We used Mean shift mainly as a diagnostic tool to cluster dispersion functions based on their similarity. Similarity is measured

by the Euclidean distance between two dispersion function vectors. This Euclidean distance does, however, not provide information on the underlying cause of dissimilarity and thus a minor difference in the values of the dispersion functions, even though characterized by a very similar shape, could result in the same level of dissimilarity as a change in the shape of the dispersion function. We observed some cluster separations that were caused by minor differences in mean dispersion, while essentially describing the same spatial dependency.

It is possible to train better Mean shift algorithm instances. As described in the methods, we selected the bandwidth parameter for Mean shift to yield meaningful results for the entire data set. The same parameter was used for all subsets to cluster dispersion functions on the same basis. This makes the clustering procedure itself comparable and thus, the number of identified clusters can support result interpretation. Nevertheless, it is likely that better bandwidth parameters can be found for each data

subset individually and overcome misclassifications as described above. Our objective, however, was to find clustering results that can directly be compared to each other (instead of comparing hyper-parameters).

Dispersion functions operate in a higher dimensional space and might be affected by the curse of dimensionality. Mean shift clusters data points based on their distance to each other. Following the theory of the curse of dimensionality, with each added dimension (of these points), the difference of maximum and minimum distance between points become less significant (Beyer et al., 1999). On the one hand, we wish to resolve dispersion functions on as many distance lag classes as possible to gain more insight on spatial dependencies. On the other hand, each additional lag class possibly decreases the performance of Mean shift (or any other clustering algorithm) and turns the results less meaningful. We calculated dispersion functions using a 30 days aggregation window and therefore end up with 335 points for Mean shift. However, despite the limited number of points and the resulting uncertainty of cluster identification the clusters identified here seem plausible.

Mean shift is sensitive for the bandwidth parameter. As described in the methods (section 2.3), the bandwidth parameter has to be specified and has direct influence on the amount of clusters formed by the algorithm. We found a suitable parameter through trial-and-error. It would be more satisfactory to infer this crucial parameter from the data or supplementary information gathered at field campaigns. However, to our knowledge there is no such method or procedure to infer bandwidth parameters for Mean shift from a data sample.

## 4.5   Limitations of the proposed method

Successful clustering does not point out spatial dependency. Mean shift can cluster functions without spatial dependency, as it uses their distance and no actual covariance between the functions. In this case the clustering is based on differences in mean, which may not even be statistically significant. The Mean shift algorithm is not meant to test clusters for statistical independence. If two groups of points are separated or not depends only on the bandwidth parameter. Therefore the centroid functions of each cluster have to be checked for their shape and the information on spatial dependency that follows from that shape.

Our approach to find suitable bins to calculate the Shannon entropy is sensitive to outliers. We decided to rather define the width of a bin instead of their number. The reasons and necessity to do so were discussed in detail in section 4.2. As a width we used the uncertainty propagated into dispersion function distances. From all distances within uncertainty margins, we used the maximum value. In cases where this maximum distance is an outlier, it will influence the whole entropy calculations. This is a limitation to our method, but an acceptable one as it is still superior to other approaches from our point of view. Choosing the maximum distance within each year or depth (or both) will yield more bins for entropy calculations and therefore a wider range of values, but it would be very hard to compare these values.

From the point of view of the monitoring network, it has to be mentioned that the analysis of the 2013 data is likely to be less reliable, as during this period of installation the number of sensors was still lower than in the following years.

Due to the sampling design and the amount of observation points, we did not systematically test for differences of forest vs pasture plots, but ran our analyses across the two land covers. The fundamentally different shapes of cluster centroids in the summer clusters and, thus, the strong effect of vegetation altering soil moisture patterns might be partly more pronounced due

to the sampling design and not easy transferable to other sites. In our opinion, we would have made the same observations with a more stratified sampling design, as this is systematic catchment behavior, but we can neither confirm nor reject this.

## 5   Conclusions

We presented a new method to identify periods of similar spatial dispersion present in a data set. While soil moisture observations might be spatially heterogeneous, spatial patterns are much more persistent in time. We found two fundamentally different states: On the one hand rainfall-driven cluster formations, usually characterized by strong relationships between dispersion and separating distance and a characteristic length roughly matching the hillslope scale. On the other hand we found clusters forming during the vegetation period. A drying and then dry soil exhibits dispersion functions which are much flatter, indicating homogeneity across space. Interestingly, these functions flatten out by minimizing the dispersion on large distance lags, which implies that dissimilarities do not increase with separating distance. We can thus see how the soil acts as a low pass filter.

While these long lasting periods of similar spatial patterns help us to understand how and when the soil is wetting or dying in an organized manner, there are possible applications beyond this. One could use the identification of clusters to stratify data based on spatial dispersion for combined modeling. Then, for example, a set of spatio-temporal geostatistical models or hydrological models applied on each period separately, might in combination return reasonable catchment responses.

Our most interesting finding is that even a few soil moisture time series bear a considerable amount of predictive information about dynamic changes of soil moisture. We argue that distributed soil moisture reflects an organized catchment state, where soil moisture variability is not random and only a small amount of observation points is necessary to capture soil moisture dynamics.

*Code and data availability.*   Major parts of the analysis are based on the scipy (Jones et al., 2001–), scikit-learn (Pedregosa et al., 2011) and scikit-gstat package (Mälicke and Schneider, 2019). All plots were generated using the matplotlib package (Hunter, 2007). The full analysis Python scripts are published on Github (https://github.com/mmaelicke/soil-moisture-dynamics-companion-code) (Mälicke, 2019).

The data is available upon request.

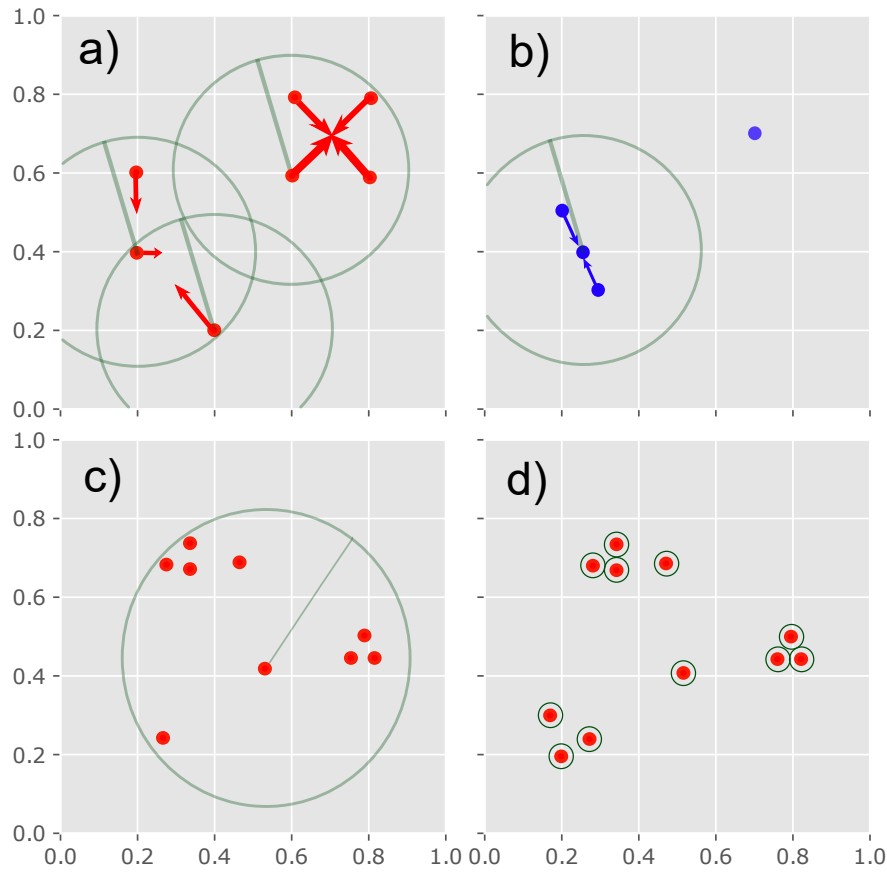

**Figure A1.** Schematic procedure of the Mean shift algorithm in $\mathbb{R}^2$. **a):** Red dots indicate hypothetical samples to be clustered. The circles are illustrating the flat kernel of the centered sample at first iteration. The bandwidth parameter is illustrated by the radius. The red arrows indicate the shift of the respective sample onto the geometric mean of all samples inside the current kernel. Note that three points on the left side are shifted differently, as the upper and lower point do not lie in each others kernel. **b):** Second iteration step after a). The blue dots are the shifted means from a) and will be used as the input sample for the next iteration. The procedure finishes when no points can be 'shifted' anymore. **c):** Example of a large bandwidth (radius), which will result in only one cluster at convergence. **d):** Example of a too small bandwidth, where no point will be shifted at all.

## Appendix A: Mean shift algorithm

Mean shift starts by forming a cluster for each sample on its own. Here, a *sample* corresponds to one dispersion function. We will illustrate the fundamental mechanism of the algorithm for the two-dimensional case, as the samples can easily be plotted in $\mathbb{R}^2$ (see figure A1 a, b). Mean shift works iteratively. In each iteration, a window is shifted over all samples, which can be thought of as coordinate points in the two-dimensional case (see fig. A1 a). This window is called a *kernel* that is controlled

by a size parameter called *bandwidth*, which is the Euclidean distance between two samples. In the two-dimensional case, this can be thought of a circle with a radius set to the given bandwidth as shown in figure A1 a. In each kernel position, the center of sample density is calculated and the current sample is *shifted* onto this point, which is the new cluster mean, called cluster centroid. On the next iteration, the newly created cluster centroids are used as the new (input) samples, as shown in figure A1 b. Hence, with the bandwidth, we define a maximum Euclidean distance at which two samples are still considered to belong to the same group. The iterations stop when the shifting means converge (centroids do not change their position anymore). We substitute the centroids calculated on the last iteration by the original sample closest to this point. Thus, we choose the most representative dispersion function for the cluster.

Mean shift is sensitive to the selected bandwidth. Two clusters whose centroids are within one bandwidth length will be shifted into a combined cluster before convergence is met. As a result a bandwidth parameter chosen too big might classify all samples as a single cluster as indicated in figure A1 c. In case the bandwidth is chosen too small many tiny clusters with just a few members will be the result. Figure A1 d shows an extreme example, where no sample will shift anywhere. We tested different bandwidth parameters at a few examples and set the bandwidth to the 30% percentile of all pairwise Euclidean vector distances between the dispersion functions of one year and depth. We chose the so-called flat kernel as a kernel, which would result in a circle in the two-dimensional case and a N-dimensional sphere in the $\mathbb{R}^N$, where $N$ is the number of lag classes used for the dispersion function.

**Table B1.** Quantitative results summary. For each depth and cluster of 2016 different cluster characteristics were calculated. The duration of each cluster is given in the third column. To compare rainfall forcing with the emergence of clusters, the rainfall characteristics were based on the same moving window as the clusters. The mean rainfall frequency $f_{30}$ within each window is given in the fourth column. The mean 30 day-sum over the whole cluster $\sum_{i=0}^{30} R$ in the fifths column. To assess the variability of dispersion functions within each cluster, different measures are given. $\gamma$ is the dispersion, as calculated in equation (2). This describes the dispersion of dispersion functions within one cluster. $H$ is the entropy of the distribution of all cluster members within each cluster. Both measures are calculated for the distribution of each distance lag class individually.

| depth | cluster | duration [days] | $f_{30}$ | $\sum_{i=0}^{30} R$ | $\gamma$ | $H$ |
|-------|---------|-----------------|----------|---------------------|----------|-----|
| 10cm | blue | 167 | 16.22 | 59.39 | 5.5e-6 | 1.34 |
| 10cm | orange | 88 | 17.48 | 73.73 | 1.12e-5 | 0.69 |
| 10cm | green | 113 | 20.54 | 81.31 | 4.36e-5 | 1.25 |
| 30cm | blue | 135 | 17.84 | 73.32 | 2.16e-5 | 1.83 |
| 30cm | orange | 136 | 15.78 | 54.87 | 4.21e-6 | 1.63 |
| 30cm | green | 42 | 21.86 | 100.62 | 4.4e-5 | 1.31 |
| 50cm | blue | 282 | 16.41 | 60.66 | 3.51e-5 | 0.99 |
| 50cm | orange | 54 | 21.59 | 98.02 | 3.64e-5 | 0.9 |

## Appendix B: Auxiliary quantitative results

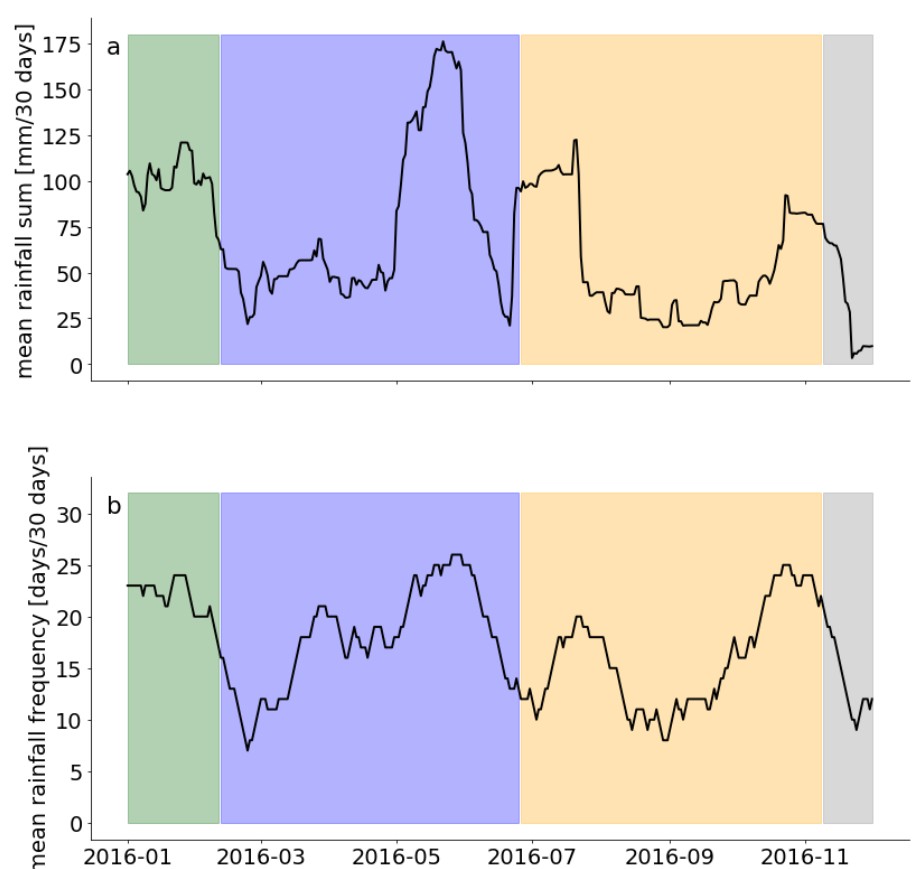

**Figure B1.** Mean rolling rainfall sum (a) and rainfall frequency (b) in for 2016. The colored boxes indicate the current cluster as shown in figure 3 d. Both values were calculated for the same windows as the dispersion functions by using equation (1) for the daily rainfall sums, with the total rainfall sum in the window in (a) and the number of days rainfall occured in (b).

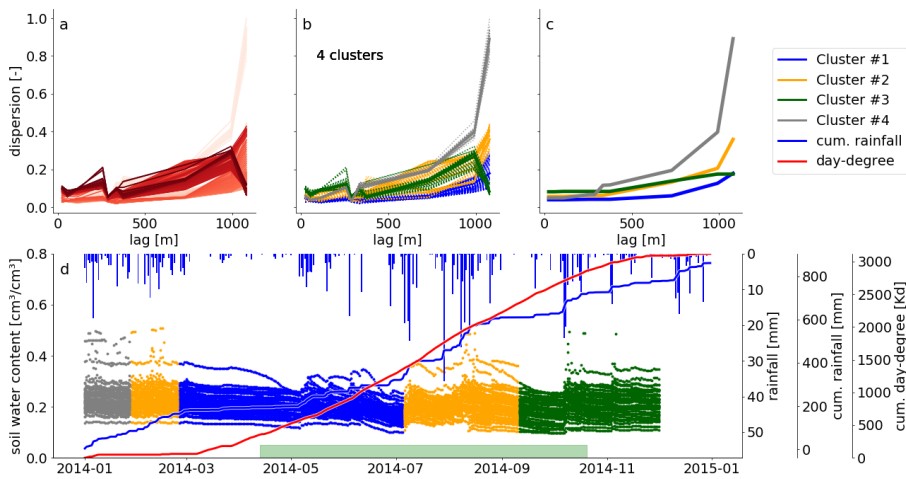

**Figure C1.** Spatial dispersion functions in 30 cm for 2014 based on on a window size of 30 days.

**a):** Spatial dispersion function for each position of the moving window. The red color saturation is indicating the window position. The darker the red the higher in the year.

**b):** The same dispersion functions as presented in a). Here the color indicates cluster membership as identified by the Mean shift algorithm.

**c):** Compressed spatial dispersion information represented by corrected cluster centroids. The colors match the clusters as presented in b).

**d):** Soil moisture time series of 2014 in 30 cm depth. The colors identify the cluster membership of the spatial dispersion function of the current window location and is matching the colors in b) and c). The bars on the top show the daily precipitation sums. The solid blue line is the cumulative daily precipitation sum and the red line the cumulative sum of all mean daily temperatures $> 5°\text{C}$. The green bar marks the assumed vegetation period. It covers the dates where the cumulative day-degree sum is $> 15\%$ & $< 90\%$ of the maximum.

## Appendix C: Detailed result plots of 30 cm in 2014 and 2015

*Author contributions.* The methodology was developed by MM, supervised by EZ and discussed with SH. The data was provided by TB and MW. All code was developed by MM. The manuscript was written by MM, with contributions of EZ in the introduction, discussion and the formulas. SH supplied the field and data descriptions. Structure, narrative and language of the manuscript was revised and significantly improved by TB.

*Competing interests.* The authors declare to have no competing interests.

*Acknowledgements.* We thank the German Ministerium für Wissenschaft, Forschung und Kunst, Baden-Württemberg for funding the V-FOR-WaTer project. We thank the German Research Foundation (DFG) for funding of the CAOS research unit FOR 1598. We especially

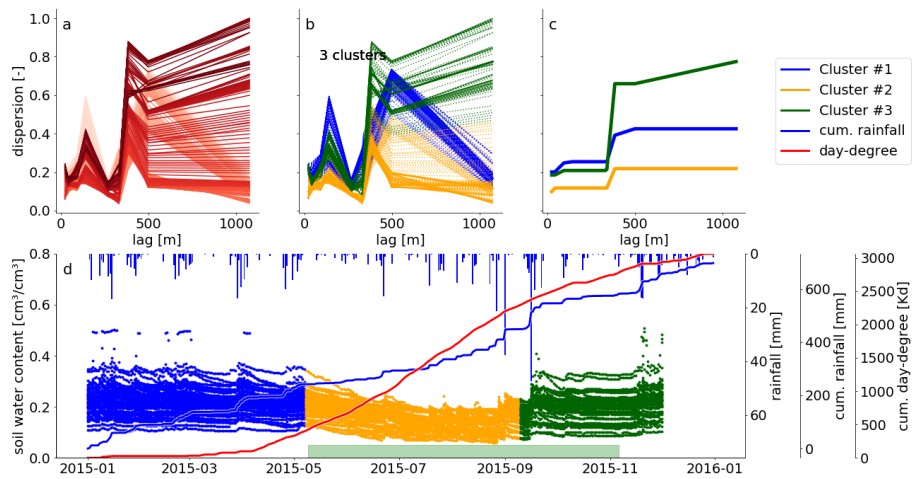

**Figure C2.** Spatial dispersion functions in 30 cm for 2015 based on on a window size of 30 days.

**a):** Spatial dispersion function for each position of the moving window. The red color saturation is indicating the window position. The darker the red the later in the year.

**b):** The same dispersion functions as presented in a). Here the color indicates cluster membership as identified by the Mean shift algorithm.

**c):** Compressed spatial dispersion information represented by corrected cluster centroids. The colors match the clusters as presented in b).

**d):** Soil moisture time series of 2015 in 30 cm depth. The colors identify the cluster membership of the spatial dispersion function of the current window location and is matching the colors in b) and c). The bars on the top show the daily precipitation sums. The solid blue line is the cumulative daily precipitation sum and the red line the cumulative sum of all mean daily temperatures $> 5°C$. The green bar marks the assumed vegetation period. It covers the dates where the cumulative day-degree sum is $> 15\% \;\&\; < 90\%$ of the maximum.

acknowledge Britta Kattenstroth and Tobias Vetter, the technicians in charge of the maintenance of the monitoring network. The authors also acknowledge support by Deutsche Forschungsgemeinschaft and the Open Access Publishing Fund of Karlsruhe Institute of Technology (KIT). The service charges for this open access publication have been covered by a Research Centre of the Helmholtz Association.

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
