# Peer review of "Soil moisture: variable in space but redundant in time"

_Hydrology and Earth System Sciences, 2019_

## Short Comment (SC1) · 10 Nov 2019

The authors present an interesting analysis of soil moisture variability. However they forget to acknowledge that many of the main results have been known for decades (like "Our most interesting finding is that even a few soil moisture time series bear a considerable amount of information about dynamic changes of soil moisture."), and some of the literature is ignored or cited in a wrong context.

The first to observe that soil moisture exhibits so-called temporal stability (persistence in the spatial pattern) were Vachaud et al (1985). Afterwards, there have been numerous (probably hundreds) of studies that have shown this persistence to be a generic feature. These studies are reviewed by Vanderlinden et al., 2012. This review should

not be referred to in the context of "Soil moisture at the headwater scale exhibits huge spatial variability and single or even distributed TDR measurements yield non-representative data", but rather to summarize the endless number of studies that have shown spatial soil moisture patterns to be persistent (such as Jacobs et al. 2004, Comegna and Basile, 1994, Grant et al.2004 among many others). It has also been argued that this pattern stability should be referred to as rank rather than temporal stability (Chen 2006).

The persistence follows directly from the water balance constraint on local soil moisture states. A powerful framework to analyse this was presented by Albertson and Montaldo (2003). Teuling and Troch used this framework to identify the spatial processes driving the persistence of soil moisture patterns and changes therein. The implications of the stability in terms of soil moisture sampling/monitoring have also been previously addressed by many other studies. The "Catchment Average Soil Moisture Monitoring (CASMM) locations" are those sites where point observations can be used to directly estimate the spatial mean (Grayson and Western 1998). We have previously shown that point-scale observations can be used to estimate the spatial mean, but due to heterogeneity in processes there is still a non-negligible error (Teuling et al 2007). More recently, Mittelbach and Seneviratne (2012) presented a framework to analysis the total soil moisture variance, and discussed the implications for spatial and temporal monitoring.

I believe the presented work needs a better positioning in the existing literature, and I hope the references presented in this comment (as well as the references in those papers) will help the authors in addressing this.

References

Albertson, J. and Montaldo, N.: Temporal dynamics of soil moisture variability: 1. The-oretical basis, Water Resour. Res., 39, 1274, doi:10.1029/2002WR001616, 2003.

Chen, Y.: Letter to the Editor on "rank stability or temporal stability", Soil Sci. Soc. Am.

J., 70, 306, doi:10.2136/sssaj2005.0290l, 2006.

Grayson, R. and Western, A.: Towards areal estimation of soil water content from point measurements: time and space stability of mean response, J. Hydrol., 207, 68–82, 1998.

Vachaud, G., Passerat De Silans, A., Balabanis, P., and Vauclin, M.: Temporal stability of spatially measured soil water probability density function, Soil Sci. Soc. Am. J., 49, 822–828, 1985

Mittelbach, H. and Seneviratne, S. I.: A new perspective on the spatio-temporal variability of soil moisture: temporal dynamics versus time-invariant contributions, Hydrol. Earth Syst. Sci., 16, 2169–2179, https://doi.org/10.5194/hess-16-2169-2012, 2012.

A. J. Teuling, R. Uijlenhoet, F. Hupet, E. E. van Loon, and P. A. Troch, Estimating spatial mean root-zone soil moisture from point-scale observations , Hydrol. Earth Syst. Sci., 10, 755–767, https://doi.org/10.5194/hess-10-755-2006, 2006.

Teuling, A. and Troch, P.: Improved understanding of soil moisture variability dynamics, Geophys. Res. Lett., 32, L05404, doi:10. 1029/2004GL021935, 2005

Jacobs, J., Mohanty, B., Hsu, E., and Miller, D.: SMEX02: Field scale variability, time stability and similarity of soil moisture, Remote Sens. Environ., 92, 436–446, doi:10.1016/j.rse.2004.02. 017, 2004.

Comegna, V. and Basile, A.: Temporal stability of spatial patterns of soil water storage in a cultivated Vesuvian soil, Geoderma, 62, 299–310, 1994.

Grant, L., Seyfried, M., and McNamara, J.: Spatial variation and temporal stability of soil water in a snow-dominated, mountain catchment, Hydrol. Processes, 18, 3493–3511, doi:10.1002/hyp.5798, 2004

---

## Short Comment (SC2) · 14 Nov 2019

I agree with the suggestion of the authors. There is definitely a need for more and new concepts dealing with spatial soil moisture variability, including the work presented in the manuscript. However the work should do justice to decades of research and insights obtained in this area.

---

## Author Comment (AC1) · 14 Nov 2019

We would like to thank Ryan Teuling for his helpful comment. His insights on temporal stability of soil water content are highly appreciated.

We admit that parts of the existing literature are not referenced in our work and we will happily review and include the given references from the short comment, wherever possible.

We agree that the review by Vanderlinden et al. (2012) should be given more visibility in our work. We will highlight the many previous studies, reviewed in Vanderlinden et al. that support our findings. However, Vanderlinden also states that the "the basic question about TS SWC [time stability of soil water content] and its controls remain

unanswered. Moreover, the evidence found in literature with respect to TS SWC controls remains contradictory." (Vanderlinden et al, 2012, p.2 l.2ff). We will stress that from our point of view the proposed method can contribute to a better understanding of if and how transitions in persistent soil moisture patterns are controlled by forcing. This is the more appropriate context to place Vanderlinden's work.

We acknowledge that a lot of work on rank stability of soil water content exists. We included only a few references on rank stability and accept that we might have to include some more references into our introduction. However, our work focuses on the methodology developed in this study. The combination of spatial dispersion functions with a clustering approach and the incorporation of uncertainty propagation into information theory to assess the loss of information is a novel approach. The calculation of information content with respect to propagated uncertainties would not be feasible using ranked data over the actual absolute values. We might have to stress this point more and the suggested references will be helpful to clarify where our method can contribute to the current understanding of soil moisture dynamics. We will therefore include a discussion of the studies by Teuling et al (2007), Mittelbach and Seneviratne (2012) and Albertson and Montaldo (2003) into our manuscript.

Albertson, J. and Montaldo, N.: Temporal dynamics of soil moisture variability: 1. Theoretical basis, Water Resour. Res., 39, 1274, doi:10.1029/2002WR001616, 2003.

Mittelbach, H. and Seneviratne, S. I.: A new perspective on the spatio-temporal variability of soil moisture: temporal dynamics versus time-invariant contributions, Hydrol. Earth Syst. Sci., 16, 2169–2179, https://doi.org/10.5194/hess-16-2169-2012, 2012.

A. J. Teuling, R. Uijlenhoet, F. Hupet, E. E. van Loon, and P. A. Troch, Estimating spatial mean root-zone soil moisture from point-scale observations , Hydrol. Earth Syst. Sci., 10, 755–767, https://doi.org/10.5194/hess-10-755-2006, 2006.

---

## Referee Comment (RC1) · Anonymous Referee #1 · 26 Nov 2019

**OVERVIEW**

The study investigates soil moisture spatial-temporal variability at hillslope and catchment scale through detailed in situ measurements for an experimental basin in Luxembourg. A new measure to capture the spatial dissimilarity is introduced. The variability of such measure in time and at different depths is computed and discussed.

**GENERAL COMMENTS**

The paper is mostly well written and clear. The topic of the paper is interesting for the readership of HESS as I believe that new statistical tools for analyzing soil moisture spatial-temporal variability are useful. Moreover, the analysis of new experimental measurements is always useful for advancing our knowledge on soil moisture variabil-

ity.

However, as highlighted by Ryan Teuling (I couldn't resist to read his comments), a large number of scientific studies introducing the concept underlined in the paper, i.e., few measurements of soil moisture can be used to characterize its temporal variability at large scale (temporal stability), have been already published. I fully agree that the paper should consider such studies more in details, it should be better located in the context of scientific literature on the topic. Therefore, several sentences and parts need to be revised.

Moreover, I believe the paper should try to add some new results for advancing our knowledge. The introduction of the new dissimilaritymeasure is one point but something more can be added (see my comments below).

On this basis, I believe the paper needs major to moderate changes before the publication; I have listed below my comments with the indication of their relevance.

1) MAJOR: Throughout the paper I have found several times only qualitative assessments, particularly in the discussion. As mentioned above, a large body of literature on soil moisture spatial-temporal variability has been published. Therefore, I believe new papers should add quantitative information that should be compared with previous studies to highlight similarities and differences. For instance, it reads that rainfall is the responsible of changes in the clustering. How much rainfall is needed to have a change in a different cluster? What is a "strong" rainfall event causing the changes? Is it rainfall amount or frequency that is important? What is the quantitative difference in spatial dissimilarities between clusters during rainingperiods? How it varies with sampling distance? How long are cluster periods in different conditions? How do they vary with depth? Some of these results can be extracted from the figures, but I believe they should be synthetized by the authors and compared (quantitatively) with results of previous studies. It is an experimental study, therefore a table highlighting the quantitative results of previous studies might be highly useful for such a comparison.

2) MODERATE: An important information that is currently missing is the "typical size of the hillslopes". Is the size of the different hillslopes similar? Can the authors add a figure with a typical configuration in 2/3 hillslopes? Moreover, what is the size of the basin?

3) MAJOR: The paper is too long in many parts (at least for me), e.g., in the description of the methods and the discussion of the results. I strongly believe the paper may benefit from a reduction of the text by focusing on the main (quantitative) results that have been obtained from the analysis of the soil moisture data. Some descriptions of the methodology can be moved in the appendix. The current version of the paper is not easy to follow.

4) MODERATE: It's not clear to me how the authors have aggregated the data for the different hillslopes. How are measurements from different hillslopes aggregated? How to address the differences in land use and topography? The problem is underlined at page 6 but not addressed in the paper.

5) MODERATE: In the discussion two "periods" are highlighted, drying and wetting. However, in the paper 3 to 4 clusters have been identified. Why? The authors should add a clear explanation for that, I believe that we do not have only drying and wetting periods, but it depends on when they occur with respect to vegetation cycle. Is it a possible explanation?

6) MODERATE: Throughout the text, some small formatting and typo corrections are needed. Please carefully check the text for such errors.

**SPECIFIC COMMENT (P: page, L: line or lines)**

P1, L10: The extent of the hillslopes and of the basins should be specified in the abstract.

P1, L14-17: The concept of "redundancy" and "compression" are clear only by reading the full paper. It is not clear by reading the abstract what is communicated here in

these sentences. Please revise.

P2, L10-11: "Soil moisture at the headwater..." This sentence should be revised, and also the paragraph L13-17.

P3, L19: The concept of "redundancy" should be clarified here in the introduction. Otherwise, it is hard to understand what the authors mean here.

P3, L28-...: The hypotheses to be tested are too specific. The authors have written such hypotheses after knowing the results (I guess). I suggest reformulating them to be less specific.

**RECOMMENDATION**

Based on the above comments, I suggest a major revision before the possible publication on Hydrology and Earth System Sciences.

---

## Author Comment (AC2) · 3 Dec 2019

We would like to thank Anonymous Referee 1 for the constructive and insightful review of our work.

**Major & Moderate comments**

In the following, we will respond to the Referee's comments in the order of appearance (Referee comments in italics). For those comments, which are in line with Ryan Teuling's (RT) comment, we refer to our corresponding response.

[Figure]

**1)** We agree that our findings need more comparison to existing literature and will revise the paper as outlined in our response to RT's comment. We thank the referee for providing various examples for presenting and interpreting our results in a more quantitative way:

**a.** *How much rainfall is needed to have a change in a different cluster?*

**Response:** As we pointed out in the manuscript, on p.17 l.32, there are instances, where a cluster transition is not forced by a rainfall event and not every significant rainfall event is causing a cluster transition. Additionally, we remind that the dispersion functions are based on a 30-days rolling mean, which makes it difficult to relate a dispersion function and therefore a cluster transition to a rainfall event on a daily basis. However, we agree that rainfall sums preceding a cluster transition might be an interesting detail to add.

**b.** *What is a "strong" rainfall event causing the changes?*

**Response:** We will define a threshold and give basic statistics on rainfall events and available soil hydraulic properties.

**c.** *Is it rainfall amount or frequency that is important?*

**Response:** This is an interesting point, we will calculate rainfall event frequency and add them as a result if it turns out to be insightful.

**d.** *What is the quantitative difference in spatial dissimilarities between clusters during raining periods? How it varies with sampling distance?*

**Response:** We think that the difference in dispersion functions are shown in various figures, and they characterize how average differences develop with increasing separating differences (similar to a semivariogram). So the spatial dissimilarities are readable from the difference on the y-axis, while the sampling distance is given on the x-axis. (e.g. Fig 4. a-c, Fig. 5 d, e, f Fig. 6 a-d).

However, we agree that a quantitative measure for the range of dispersion functions within one cluster might be helpful. We will try to find a good

measure for quantifying this range

**e.** *How long are cluster periods in different conditions?*
**Response:** Thanks for this good idea. We will extract the period lengths from the figures and give them as numbers. The fact that under dry conditions generally less cluster emerge (which last longer) is from our point of view already discussed in section 4.3 of the manuscript.

**f.** *How do they vary with depth?*
**Response:** We will extract the difference in period length over depth from Fig. 5.

**g** We agree that calculating the quantitative measures described above can be helpful to back up our findings where applicable. The Referee requests a comparison to existing quantitative results from literature. We will review the literature and include comparisons in the discussion.
We would like to emphasize that we consider qualitative descriptions of our results necessary to better understand the findings of this study. We agree that quantitative results will add clarity and comparability. Hence we will include quantitative measures as indicated in a-f in an additional result table.

**2)** *An important information that is currently missing is the "typical size of the hillslopes". Is the size of the different hillslopes similar? Can the authors add a figure with a typical configuration in 2/3 hillslopes? Moreover, what is the size of the basin?*
**Response:** We completely agree with the Referee and thank for pointing out this missing information. We will add the basin size and characterize the hillslopes in the study site description.

**3)** *The paper is too long in many parts (at least for me), e.g., in the description of the methods and the discussion of the results. I strongly believe the paper may benefit from a reduction of the text by focusing on the main (quantitative) results that*

*have been obtained from the analysis of the soil moisture data. Some descriptions of the methodology can be moved in the appendix. The current version of the paper is not easy to follow.*

**Response:** We will consider revising the methods to shorten them. However, as the development of the methodology is a major focus of this study, a detailed description of the methods is vital to understand the results.

We will revise the discussion of our results including the results that will be obtained from our comment 1).

The Referee argues that we should shorten the discussion of our results in favor of a focus on quantitative results. In comment 1) the Referee requests a more detailed comparison of quantitative results, which will be added, and comparison to the existing literature which will also be added. This is important. A detailed discussion of the already presented results is however of equal importance. We will revise the discussion to achieve a good balance between these issues and will make an effort to improve readability and ease of comprehension.

4) *It's not clear to me how the authors have aggregated the data for the different hillslopes. How are measurements from different hillslopes aggregated? How to address the differences in land use and topography? The problem is underlined at page 6 but not addressed in the paper.*

**Response:** To clarify: The data has not been pooled into datasets per hillslope but is analyzed across the whole study area. This aggregation refers to an average value for each time series within the moving window, not a spatial aggregation. See equation (1) (p.5) We will clarify the paragraph.

We hope that we understood the Referee correctly, but differences in land use and topography are addressed in the paper. The paragraph on p.6 (l.16-19) is further referencing the paragraph from l.18 to l.30 on page 8 and both are dedicated to this issue. To clarify: We only formed point pairs, which are not further apart than 1200 m, because beyond this distance the observations are most likely

located on different hillslopes. In these cases, observations might be more similar due to land use or topography, and not separating distance. We will revise the paragraphs to make this clearer.

**5)** *In the discussion two "periods" are highlighted, drying and wetting. However, in the paper 3 to 4 clusters have been identified. Why? The authors should add a clear explanation for that, I believe that we do not have only drying and wetting periods, but it depends on when they occur with respect to vegetation cycle. Is it a possible explanation?*

**Response:** We will add a clear explanation about the emergence of clusters. We will highlight that the number of identified clusters is already an important result. We will clarify in the discussion that the clusters found in winter/spring and summer/autumn are similar to each other and this broad grouping is comparable to the often cited two 'soil moisture states', that were found in the Tarrawarra catchment in Australia (Grayson et al., 1997, Western et al., 1999). We will rephrase these parts.

We agree with the Referee that there are more than only 'radiation driven drying' and 'rainfall driven wetting' periods, there is also seepage, which we loosely subsumed under drying. We also strongly agree the annual climate cycle and vegetation cycle are important seasonal controls, which might manifest in clusters. The proposed method is feasible to characterize particularly the latter influence as we use 30-days mean values. We will add a clear explanation that our method did identify more than two clusters. However, in many cases the found clusters were similar during 'wet' and 'dry' conditions (similar to Grayson). We agree with the Referee that we should directly address this in the manuscript. We will clarify the respective sections.

**6)** *Throughout the text, some small formatting and typo corrections are needed. Please carefully check the text for such errors.*

**Response:** We apologize and will carefully proofread our manuscript before resubmission.

**Specific Comments:**

*P1, L10: The extent of the hillslopes and of the basins should be specified in the abstract.*
**Response:** We will add these numbers.

*P1, L14-17: The concept of "redundancy" and "compression" are clear only by reading the full paper. It is not clear by reading the abstract what is communicated here in these sentences. Please revise.*
**Response:** We will clarify these sentences.

*P2, L10-11: "Soil moisture at the headwater..." This sentence should be revised, and also the paragraph L13-17*
**Response:** As indicated in our response to RT's comment, we will revise this part and correct wrong contextualized references.

*P3, L19: The concept of "redundancy" should be clarified here in the introduction. Otherwise, it is hard to understand what the authors mean here.*
**Response:** We will clarify and add an explanation on 'redundancy'

*P3, L28-...: The hypotheses to be tested are too specific. The authors have written such hypotheses after knowing the results (I guess). I suggest reformulating them to be less specific*
**Response:** We respectfully note that we think our hypotheses should be specific to be testable. Each of them is focusing on one part of the proposed method to guide the reader through our manuscript. However, we suggest a revision of the paragraphs in the introduction, which leads to our hypothesis. This will help to better follow our argumentation. The affected paragraphs include: H1: p.2 L.18-21; H2: p.3 L.10-15;

[Figure]

H3: p.2 L. 23-28; H4: p.3 L.18-24.

Finally, we would like to thank the Referee for his helpful, detailed and insightful review of our work.

**References**

Grayson, R. B., Western, A. W., Chiew, F. H. S., Blöschl, G. (1997). Preferred states in spatial soil moisture patterns: Local and nonlocal controls. *Water Resources Research, 33*(12), 2897–2908. https://doi.org/10.1029/97WR02174

Western, A. W., Grayson, R. B., Blöschl, G., Willgoose, G. R., McMahon, T. A. (1999). Observed spatial organization of soil moisture and its relation to terrain indices. *Water Resources Research, 35*(3), 797–810. https://doi.org/10.1029/1998WR900065

---

## Referee Comment (RC2) · Anonymous Referee #2 · 4 Dec 2019

I am not a hydrologist, so I cannot say anything about the level of novelty of the current work with respect to the published literature of which I am not well aware. On the other hand, the proposed methodology seems very reasonable and effective to me. I liked the paper and the interpretation of the results. On the other hand, I personally found that the text should be improved: some sentences are open to ambiguity or unclear (the meaning or idea to be conveyed is there intuitively, but the structure of the sentence leaves it open to misinterpretations), and there are some repetitions that could be cut out in order to make the paper easier to follow (sometimes is hard). I suppose that the confusion in some sentence stem from the young age of the first Author: put some more efforts in making the text clearer and more specific in order to honor your work.

Comment 1: pp 9, lines 26-27: 'This distance is the foundation of Mean shift on the one

hand and the compression quality calculations presented in section 2.5.2 below on the other hand'. The relevance of the distance in (3) for the mean shift clustering algorithm is somehow hidden during the description of the method in Sec. 2.3.1 (i.e., 'We tested different bandwidth parameters at a few examples and set the bandwidth to the 30% percentile of all pairwise vector distances between the dispersion functions of one year and depth'). If possible, I would like to rephrase this aspect putting more focus on the relevance of (3) within the clustering algorithm.

Comment 2: In the Section 2.5.2 Compression quality, there are some unclear aspects to me. 'To assure comparability we use one binning for all calculations of H (across years and depths). To achieve this, all pairwise distances between all spatial dispersion functions of all four years in all three depths are calculated. The discrete frequency distribution is formed from 0 up to the global maximum distance (between two dispersion functions) calculated using equation (3). The bins are formed equidistant using a width of the maximum function distance that still lies within the error margins calculated using equation (9). Thus, the information content of the spatial heterogeneity is calculated with respect to the expected uncertainties. This way we can be sure to distinguish exclusively those spatial dispersion functions that lie outside the error margins.' It is my understanding that the binning scheme is grounded on the distance function in (3), which make me to think that subsequent calculation (e.g., entropy, KL-divergence) will involve the distance in (3), but then 'To calculate the mean information content of the compressed series each cluster member is substituted by the respective cluster centroid. This substitution is obviously not a compression in a technical sense, but necessary to calculate the Kullback-Leibler divergence. Then a frequency distribution for compressed series X and the uncompressed series Y can be calculated. The Kullback-Leibler divergence DKL of X,Y is given in equation (12)' which compare compressed and uncompressed dispersion functions and does not involve the distance in (3) in any way. It seems that the binning (size of the bin and edges of the diverse bins) entails the diverse distance according to (3) (and the associated uncertainty, according with (9)), but the KL is evaluated for the dispersion function themselves (and not their

distance)? What am I missing? Could you please further clarify how the distance (3) is involved in the evaluation of (12)? I would also briefly describe the meaning of the KL-divergence which is just introduced, but not commented.

Comment 3: pp.12 lines 6-7: 'Dispersion declines with separating distance, as small values correspond to observations which have similar values while large values suggest the opposite' looking at Fig. 4 the dispersion function increases with the spatial lag, such that dispersion increases with separating distance. What am I missing here?

Comment 4: pp.12 lines 18-23: 'As the spatial dispersion functions in the presented example are redundant in time, we compressed the information by replacing the dispersion function within one cluster by the cluster centroid. All four representative functions shown in Figure 4 c) exhibit increasing dispersion with separating distance. For the blue and green cluster this happens step-wise at a characteristic distance of 500 m. That reminds us of a Gaussian variogram, which can also show a step-wise characteristic. The small grey cluster shows an increase at 500 and another one at 1000 m separating distance. In contrast the orange cluster, however, shows a only a gentle increase with distance.' Are there any reasons for these stepwise increases or some physical related explanations for these behaviors? Comment 5: with reference to Fig. 4 and its discussion in Sec. 3.1 vegetation period is mentioned several times, would it be nice to have on Fig. 4.c this period highlighted (also in other figures where vegetation period is of relevance), for example as a light green bar along the x-axe or similarly, in order to understand when this vegetation period is 'on/off'.

Comment 6: pp. 12, lines 24-26: 'In the vegetation period observations are similar even at large separating distances. As the orange cluster (Figure 4 c and d) covers significant parts of the vegetation period, the influence of vegetation on spatial soil water dynamics is considered to be dominant'. The latter sentence is misleading: during the so referred vegetation period which are the concurrent factors along with the vegetation-related influence that could possibly influence the soil moisture? How is it possible to discern the impacts of other factors in order to say that vegetation influence

is the dominant one? Or, if the vegetation-related influence is the only factor, no surprise that it is the dominant one. Please consider revise the sentence or better support it.

Comment 7: pp.13, lines 7-8: 'At the same time the observations get spatially more homogeneous in summer, particularly when the blue cluster emerges, because the dispersion at large lags decreases significantly.' I would substitute 'because' with 'i.e.', the decrease of the dispersion function is a consequence not a cause of the more homogeneous nature of the data during summer.

Comment 8: pp. 13 lines 13-15: 'The green clusters emerge with strong rainfall events after longer previous dry spells (Fig. 5). We would have expected a third occurrence at the beginning of August, but the soil may already be too dry to bear a detectable dependency on separating distance'. It is not clear at which green cluster the Authors are referring to in this sentence, please clarify.

Comment 9: pp. 13 lines 17-18: 'The 50 cm dispersion functions (Fig. 5 f) show a clear spatial dependence and are similar'. I can see the trends of the two dispersion functions with respect to the lag and I can see that these trends are consistent among each other, but I don't see the similarity between the two dispersion functions that are characterized by fairly different values especially at larger lag. Please revise the sentence having care of the specificity of the wording.

Comment 10: pp. 13 lines 18-19: 'Not only the soil moisture observations have become much more homogeneous with depth, also the dispersion functions are more similar in shape.' The first part of the sentence is vague, more homogeneous with respect to what? In time or space? Looking at Fig. 5c, at fixed time (e.g., 2016-02), I can see a great level of heterogeneity across the moisture data, even larger than that recorded in 5.a-b. Furthermore, the dispersion function orange in Fig. 5f reaches values comparable to that of the dispersion functions in Fig. 5d (orange) and Fig. 5e (grey and green clusters).

[Figure]

Comment 11: pp. 14 lines 1-2: 'We find dispersion functions with characteristic length of 500 m and the blue cluster persists throughout most of the year.' How is the characteristic length of the dispersion function defined? Please clarify.

Comment 12: pp. 16 lines 4: 'water water dynamics', water is repeated.

Comment 13: pp. 18 line 27: 'In line with H4 spatial patterns of soil moisture were found to be persistent over longer time periods' Longer than what?

Comment 14: pp. 19 lines 17-18: 'We thus conclude that there is dependence of the dispersion on the rainfall pattern, which is reflected in their shape and characteristic lengths.' The sentence, as written, means that the shape and characteristic lengths are referred to that of the rainfall pattern, while I am imagining that are the shape and characteristic lengths of the dispersion function the ones that changes. Please revise the sentence.

Comment 15: pp 20 lines 16-18: 'This Euclidean distance does, however, not provide information on the underlying cause of dissimilarity and thus a simple shift along the y-axis can result in the same level of dissimilarity as a change in the shape of the dispersion function.' Despite being clear from an intuitive point of view, this sentence can be sloppy to the most rigorous reader: the y-axe of what? Please revise, like 'a minor difference in the values of the dispersion functions, even though characterized by the a very similar shape, could results in . . . .'.

Comment 16: pp 22 line 1: 'A drying and then dry soil exhibits dispersion functions without spatial structure. Interestingly, these functions flatten out by minimizing the dispersion on large distance lags and we can thus see how the soil acts as a low pass filter.' The first sentence is obscure, especially when linked with the second one. Why the Authors claim that there is no spatial structure during drying and dry periods, when the associated dispersion functions clearly show a flat behavior for the majority of the spatial lags? As far as I have understood, the latter behavior is a sign of homogeneity in the soil moisture across space, which is a clear sign of a structure in space (maybe

not that interesting, though) to me.

---

## Author Comment (AC3) · 9 Dec 2019

We would like to thank Anonymous Referee 2 for the very constructive and insightful review of our work.

In the following, we will respond to the Referee's comments in the order of appearance (Referee comments in italics).

General comment:

*I am not a hydrologist, so I cannot say anything about the level of novelty of the current work with respect to the published literature of which I am not well aware. On the*

[Figure]

*other hand, the proposed methodology seems very reasonable and effective to me. I liked the paper and the interpretation of the results. On the other hand, I personally found that the text should be improved: some sentences are open to ambiguity or unclear (the meaning or idea to be conveyed is there intuitively, but the structure of the sentence leaves it open to misinterpretations), and there are some repetitions that could be cutout in order to make the paper easier to follow (sometimes is hard). I suppose that the confusion in some sentence stem from the young age of the first Author: put some more efforts in making the text clearer and more specific in order to honor your work.*

**Response:** We are very happy to hear the reviewer liked our manuscript and we will make an effort to improve the writing to reduce ambiguities and repetitions.

Specific comments:

**1)** *pp 9, lines 26-27: 'This distance is the foundation of Mean shift on the one hand and the compression quality calculations presented in section 2.5.2 below on the other hand'. The relevance of the distance in (3) for the mean shift clustering algorithm is somehow hidden during the description of the method in Sec. 2.3.1 (i.e., 'We tested different bandwidth parameters at a few examples and set the bandwidth to the 30% percentile of all pairwise vector distances between the dispersion functions of one year and depth'). If possible, I would like to rephrase this aspect putting more focus on the relevance of (3) within the clustering algorithm.*

**Response:** We will rephrase the respective paragraph. Equation (3) is important for Mean shift, as distances between input features in Mean shift are also calculated using the Euclidean distance. There are other algorithms that are i.e. based on ranks. We will put more focus on the relevance of Euclidean distance for Mean shift.

**2)** *In the Section 2.5.2 Compression quality, there are some unclear aspects to me.*

[Figure]

*'To assure comparability we use one binning for all calculations of H (across years and depths). To achieve this, all pairwise distances between all spatial dispersion functions of all four years in all three depths are calculated. The discrete frequency distribution is formed from 0 up to the global maximum distance (between two dispersion functions) calculated using equation (3). The bins are formed equidistant using a width of the maximum function distance that still lies within the error margins calculated using equation (9). Thus, the information content of the spatial heterogeneity is calculated with respect to the expected uncertainties. This way we can be sure to distinguish exclusively those spatial dispersion functions that lie outside the error margins.' It is my understanding that the binning scheme is grounded on the distance function in (3), which make me to think that subsequent calculation (e.g., entropy, KLdivergence) will involve the distance in (3), but then 'To calculate the mean information content of the compressed series each cluster member is substituted by the respective cluster centroid. This substitution is obviously not a compression in a technical sense, but necessary to calculate the Kullback-Leibler divergence. Then a frequency distribution for compressed series X and the uncompressed series Y can be calculated. The Kullback-Leibler divergence DKL of X,Y is given in equation (12)' which compare compressed and uncompressed dispersion functions and does not involve the distance in (3) in any way. It seems that the binning (size of the bin and edges of the diverse bins) entails the diverse distance according to (3) (and the associated uncertainty, according with (9)), but the KL is evaluated for the dispersion function themselves (and not their distance)? What am I missing? Could you please further clarify how the distance (3) is involved in the evaluation of (12)? I would also briefly describe the meaning of the KL-divergence which is just introduced, but not commented.*

**Response:** We thank the referee for pointing this out. We agree that the Kullback-Leibler divergence (KL) should be further described in section 2.5.2 (p.11). The Referee is right, this part can lead to misunderstandings and we

will carefully clarify respective sections. The main reason for possible confusion is that we combine two methods that both involve a step called binning. These two binnings are not linked to each other. The first binning refers to the dispersion function, here we pool observation points into lag distance classes for calculating the dispersion function. In the revised manuscript we will strictly introduce this binning as lag classes.

The second binning is necessary for calculating the Shannon entropy (equation 11) and therefore also the KL, which is using Shannon entropy. To this end we have to treat the set of distances between all dispersion functions as a discrete pdf. Thus "all pairwise distances between all spatial dispersion functions of all four years in all three depths are calculated" need to be pooled into meaningful distance classes (p.10 L22-23). Following Loritz et al. 2018 we think a meaningful minimum distance should be larger than the error margin. So, yes we base the calculation of KL on the distances between dispersion functions and not the dispersion functions themselves. And, the Referee is right, the binning for equation 12 is not the same as for calculating the dispersion function. We will clarify this in the revised manuscript.

3) *pp.12 lines 6-7: 'Dispersion declines with separating distance, as small values correspond to observations which have similar values while large values suggest the opposite' looking at Fig. 4 the dispersion function increases with the spatial lag, such that dispersion increases with separating distance. What am I missing here?*
**Response:** The Referee is right, this is a mistake. The dispersion is increasing with distance. We will replace 'decline' with 'increases'.

4) *pp.12 lines 18-23: 'As the spatial dispersion functions in the presented example are redundant in time, we compressed the information by replacing the dispersion function within one cluster by the cluster centroid. All four representative functions shown in Figure 4 c) exhibit increasing dispersion with separating dis-*

*tance. For the blue and green cluster this happens step-wise at a characteristic distance of 500 m. That reminds us of a Gaussian variogram, which can also show a step-wise characteristic. The small grey cluster shows an increase at 500 and another one at 1000 m separating distance. In contrast the orange cluster, however, shows only a gentle increase with distance.' Are there any reasons for these stepwise increases or some physical related explanations for these behaviors?*

**Response:** In geostatistics we may fit the Gaussian variogram function to experimental variograms which have strongly changing slopes. In close proximity to an observation it is very similar and the semivariance is rather constant, this is followed by a strong increase over a rather short increase in the lag distance. An experimental variogram of a variable that is closely linked to land use might look like this, if land use changes significantly within short distances. In my opinion, as this can only be seen in winter and spring, this originates from heterogeneous rainfall/throughfall input, whose spatial structure is still present in the dispersion function. Within the vegetation period these structures get evened out by the dominating effect of transpiring vegetation taking up considerable amounts of water before this water can shape soil moisture patterns. However, we cannot provide evidence that this explanation is correct.

5) *with reference to Fig. 4 and its discussion in Sec. 3.1 vegetation period is mentioned several times, would it be nice to have on Fig. 4.c this period highlighted (also in other figures where vegetation period is of relevance), for example as a light green bar along the x-axe or similarly, in order to understand when this vegetation period is 'on/off'.*

**Response:** Thank you for this good idea. We will add the suggested bar after defining a means for inferring vegetation period from temperature-sum curves.

6) *pp. 12, lines 24-26: 'In the vegetation period observations are similar even at large separating distances. As the orange cluster (Figure 4 c and d) covers significant*

*parts of the vegetation period, the influence of vegetation on spatial soil water dynamics is considered to be dominant'. The latter sentence is misleading: during the so referred vegetation period which are the concurrent factors along with the vegetation-related influence that could possibly influence the soil moisture? How is it possible to discern the impacts of other factors in order to say that vegetation influence is the dominant one? Or, if the vegetation-related influence is the only factor, no surprise that it is the dominant one. Please consider revise the sentence or better support it.*

**Response:** We agree with the Referee. We derived the 'vegetation period' only from the temperature-sum curve, following an approach equivalent to e.g Solantie (2004) or Seibert et al. (2017), and therefore cannot discern impacts of other factors. We will revise the paragraph to better express that dispersion functions are fundamentally different in summer/autumn (and how they are different). That the vegetation is responsible, is one possible explanation and we will move this part to the discussion.

**7)** *pp.13, lines 7-8: 'At the same time the observations get spatially more homogeneous in summer, particularly when the blue cluster emerges, because the dispersion at large lags decreases significantly.' I would substitute 'because' with 'i.e.', the decrease of the dispersion function is a consequence not a cause of the more homogeneous nature of the data during summer.*

**Response:** The Referee is right and we will change the sentence as suggested.

**8)** *pp. 13 lines 13-15: 'The green clusters emerge with strong rainfall events after longer previous dry spells (Fig. 5). We would have expected a third occurrence at the beginning of August, but the soil may already be too dry to bear a detectable dependency on separating distance'. It is not clear at which green cluster the Authors are referring to in this sentence, please clarify.*

**Response:** Thank you for pointing this out. We will reference Fig. 5 a,d instead of just Fig. 5.

**9)** *pp. 13 lines 17-18: 'The 50 cm dispersion functions (Fig. 5 f) show a clear spatial dependence and are similar'. I can see the trends of the two dispersion functions with respect to the lag and I can see that these trends are consistent among each other, but I don't see the similarity between the two dispersion functions that are characterized by fairly different values especially at larger lag. Please revise the sentence having care of the specificity of the wording.*
**Response:** We were intending to point out that the dispersion value at large lags is the only difference and will rephrase the sentence accordingly.

**10)** *pp. 13 lines 18-19: 'Not only the soil moisture observations have become much more homogeneous with depth, also the dispersion functions are more similar in shape.' The first part of the sentence is vague, more homogeneous with respect to what? In time or space? Looking at Fig. 5c, at fixed time (e.g., 2016-02), I can see a great level of heterogeneity across the moisture data, even larger than that recorded in 5.a-b. Furthermore, the dispersion function orange in Fig. 5f reaches values comparable to that of the dispersion functions in Fig. 5d (orange) and Fig. 5e (grey and green clusters).*
**Response:** We agree that our description of figure 5 needs to be specified. We will put more emphasis on the fact that in Fig. 5f the two functions are of similar shape, and only differ in the value of the last few lag classes. In contrast, 5d and 5e show dispersion functions of different shape. In Fig. 5d we find increasing functions vs. a flat function and in 5e two step-wise functions vs an increasing function. We agree that this has to be explained more properly. Furthermore we realized that the statement 'soil moisture observations have become much more homogeneous' does not hold like this will revise it.

**11)** *pp. 14 lines 1-2: 'We find dispersion functions with characteristic length of 500 m and the blue cluster persists throughout most of the year.' How is the characteristic length of the dispersion function defined? Please clarify.*
**Response:** With 'characteristic length' we are referring to the correlation length

of the dispersion function. We will add a definition to the respective section in the methods, where dispersion functions are introduced (2.2, p. 5 -6).

**12)** *pp. 16 lines 4: 'water water dynamics', water is repeated.*
**Response:** Thank you.

**13)** *pp. 18 line 27: 'In line with H4 spatial patterns of soil moisture were found to be persistent over longer time periods' Longer than what?*
**Response:** We did not compare the period lengths with anything specific and will therefore change the wording to '. . . were found to be persistent over weeks, if not months'.

**14)** *pp. 19 lines 17-18: 'We thus conclude that there is dependence of the dispersion on the rainfall pattern, which is reflected in their shape and characteristic lengths.' The sentence, as written, means that the shape and characteristic lengths are referred to that of the rainfall pattern, while I am imagining that are the shape and characteristic lengths of the dispersion function the ones that changes. Please revise the sentence.*
**Response:** Thank you for pointing this out. The Referee is right, we are referring to the shape and characteristic lengths of the dispersion function and will therefore revise the sentence.

**15)** *pp 20 lines 16-18: 'This Euclidean distance does, however, not provide information on the underlying cause of dissimilarity and thus a simple shift along the y-axis can result in the same level of dissimilarity as a change in the shape of the dispersion function.' Despite being clear from an intuitive point of view, this sentence can be sloppy to the most rigorous reader: the y-axe of what? Please revise, like 'a minor difference in the values of the dispersion functions, even though characterized by the a very similar shape, could results in . . ..'.*
**Response:** We thank the Referee and will revise the sentence as suggested.

**16)** *pp 22 line 1: 'A drying and then dry soil exhibits dispersion functions without spatial structure. Interestingly, these functions flatten out by minimizing the dispersion on large distance lags and we can thus see how the soil acts as a low pass filter.' The first sentence is obscure, especially when linked with the second one. Why the Authors claim that there is no spatial structure during drying and dry periods, when the associated dispersion functions clearly show a flat behavior for the majority of the spatial lags? As far as I have understood, the latter behavior is a sign of homogeneity in the soil moisture across space, which is a clear sign of a structure in space (maybe not that interesting, though) to me.*

**Response:** We agree with the Referee. Homogeneity across space is also to our understanding a sign of structure. We will revise the sentence and highlight the difference to the dispersion functions in wet periods.

Finally we would like to thank the Referee for the detailed, constructive and insightful comments.

References

Loritz, R., Gupta, H., Jackisch, C., Westhoff, M., Kleidon, A., Ehret, U., and Zehe, E. (2018). On the dynamic nature of hydrological similarity. *Hydrology and Earth System Sciences, 22*, 3663–3684.

Seibert, S. P., Jackisch, C., Ehret, U., Pfister, L., Zehe, E. (2017). Unravelling abiotic and biotic controls on the seasonal water balance using data-driven dimensionless diagnostics. *Hydrology and Earth System Sciences, 21*(6), 2817-2841.

Solantie, R. (2004). Daytime temperature sum-a new thermal variable describing growing season characteristics and explaining evapotranspiration. *Boreal environment research, 9*(4), 319-334.

[Figure]

Interactive
comment